# A Computational Framework for Evaluating Human-likeness in LLMs' Open-ended Human Behaviors

**Yuxuan Lei** [1]  **Jianxun Lian** [2]  **Defu Lian** [1]  **Jincenzi Wu** [3]  **Tianfu Wang** [4]  **Xing Xie** [2]

## Abstract

Large Language Models (LLMs) have found widespread application and research in scenarios such as role-playing and sociological simulations. Despite the growing use of LLM-based agents to simulate human activities, the extent to which their behaviors resemble human behavior remains underexplored. As diverse LLMs proliferate, the traditional Turing test is ineffective for scalable evaluation and prone to bias from human-crafted challenges, leading to unfair assessments. In this work, we propose a novel distribution-based framework that comprehensively evaluates human-likeness and believability of AI behaviors by leveraging large-scale open-ended human behavior data from web. For better evaluation, we design generic metrics to cover three principles: rationality, consistency, and diversity. Implemented across online shopping, open-topic Q&A, and urban mobility, our framework reveals that even the currently best LLM still exhibits a significant gap from real user behavior, underscoring the necessity of comprehensive research and evaluation of AI's human-like capabilities.

## 1. Introduction

In recent years, Large Language Models (LLMs) have achieved remarkable breakthroughs, often reaching human-level performance on professional benchmarks (Ahn et al., 2024; Jiang et al., 2026). Beyond serving as helpful assistants, they are increasingly used to simulate human behavior, enabling applications like intelligent NPCs (Wang et al., 2025b), emotional companions (Xu et al., 2024), behav-

[1]University of Science and Technology of China, Hefei, China [2]Microsoft Research Asia, Beijing, China [3]The Chinese University of Hong Kong, Hong Kong, China [4]The Hong Kong University of Science and Technology (Guangzhou), Guangzhou, China. Correspondence to: Jianxun Lian <jialia@microsoft.com>.

*Proceedings of the 43rd International Conference on Machine Learning*, Seoul, South Korea. PMLR 306, 2026. Copyright 2026 by the author(s).

ior synthesis for scientific discovery (Gürcan, 2024), and customer behavior simulation (Kasuga & Yonetani, 2024). While LLMs can generate vivid human-like behaviors (Park et al., 2023), their believability remains an open question. How closely do AI-generated behaviors resemble real human actions? Can they capture real-world population patterns? A rigorous investigation is crucial, as unqualified LLM-based simulations risk leading to misleading conclusions in fields like social science.

The Turing test (Turing, 1987) evaluates AI human-likeness by assessing whether humans can distinguish AI from real people. Recent studies (Jones et al., 2025; Jannai et al., 2023) show that powerful AIs like GPT-4 can pass this test. However, it has key limitations: (1) it focuses solely on conversation within short timeframes (2 minutes in (Jannai et al., 2023) and 5 minutes in (Jones et al., 2025)), capturing only a fraction of real-world behaviors, and (2) human participants, aware of the test, may adopt adversarial strategies, deviating from reflecting daily behaviors. Other approaches, such as using economic games or psychological assessments (Mei et al., 2024; Hu et al., 2026) to evaluate the human-likeness of AI behaviors, avoid recruiting human participants for direct human-AI comparisons but risk LLMs memorizing well-known behavioral or psychological tests during pre-training rather than generating spontaneous responses. More importantly, these works focus on single-step tasks with a unique correct answer, implicitly treating human behavior as objective and deterministic, and thus fail to capture the subjective and distributional nature present in many human decision-making processes.

Motivated by this, we propose a computational framework for human-likeness evaluation using large-scale web data. Datasets such as Amazon for online shopping, Stackexchange for open-topic Q&A, Foursquare for urban mobility, capture open-ended human behaviors in the sense that multiple actions are plausible under the same context. These datasets go beyond conversations and capture sequential human actions, minimizing bias from adversarial intent and reducing the risk of AIs echoing memorized behaviors. Additionally, our framework quantifies the similarity between AI-generated and real human behaviors at the distribution level, rather than against a single objective label. By mod-

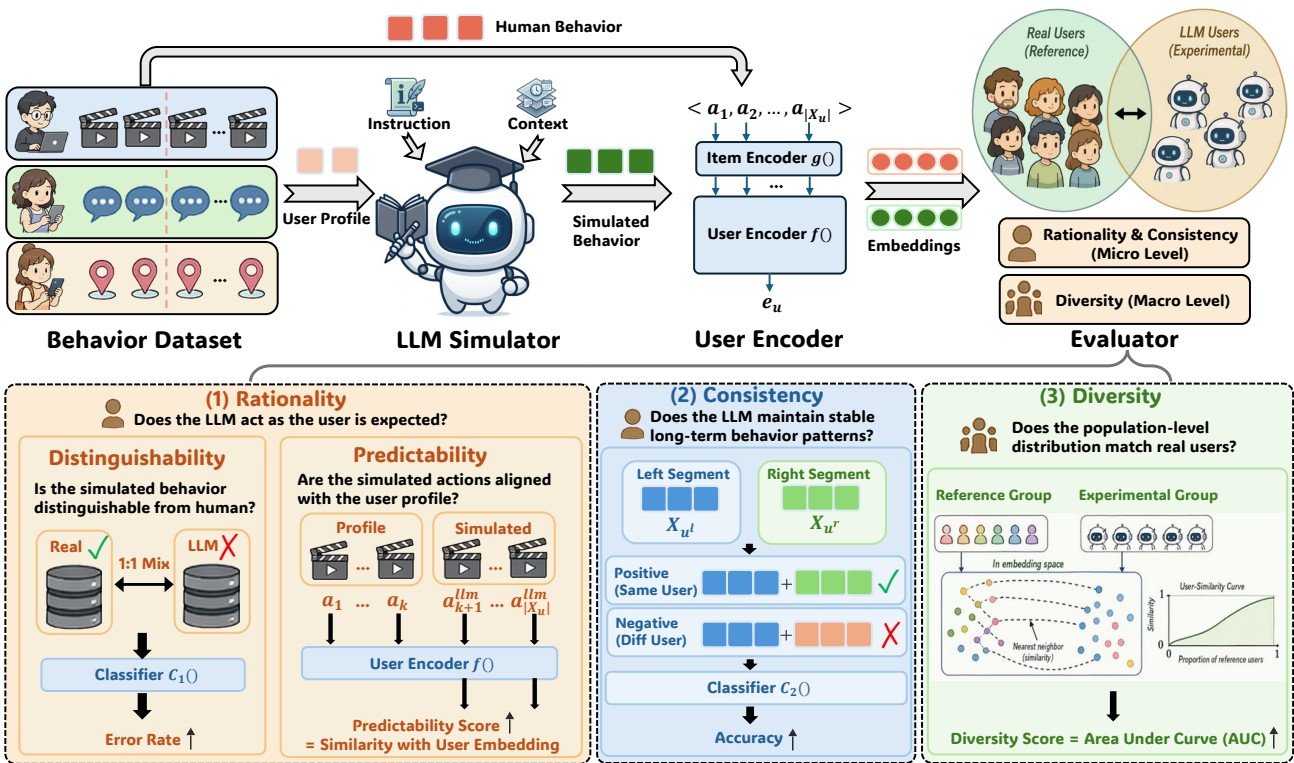

Figure 1. A graphical illustration for our evaluation framework.

eling the heterogeneous and stochastic patterns of human behavior over sequences, it enables robust, sequence-aware evaluations that go beyond the limitations of single-step, correctness-based benchmarks. To measure both the human-likeness and believability of AI behaviors, we design generic metrics to cover three principles: rationality, consistency, and diversity. Rationality ensures instant human-likeness, measured by both distinguishability and a new predictability metric. However, instant human-likeness alone is insufficient—AI must maintain consistency across behavior sequence to avoid irrational shifts, like divergent personalities. Diversity measures the capability of AI to reflect the spectrum of the human population. This principle is particularly important for social simulation, where heterogeneous AI behaviors are crucial for drawing meaningful conclusions from the simulation.

To verify the feasibility and generalizability of our framework, we extract three behavior datasets from the Web: Amazon for online shopping, Stackexchange for open-topic Q&A, and Foursquare check-ins for urban mobility. The shopping dataset captures user preferences, the Q&A dataset focuses on content generation and opinions, and the mobility dataset reflects movement patterns. We evaluate mainstream LLMs on them, revealing significant differences in their ability to simulate human behavior. More advanced models exhibit stronger human-like capabilities, with GPT-4o leading, though still falling short of real users. Our work highlights

the importance of evaluating and improving AI's human-like abilities and serves as a general evaluation framework, paving the way for future research in humanoid AI.

**Contributions**. (1) We propose a large-scale, fully automated, distribution-based computational framework for evaluating the human-likeness of LLMs, leveraging public web data as a human reference and adaptable to various open-ended behaviors. (2) To assess human-likeness and believability, we introduce metrics based on three key principles: rationality, consistency, and diversity, extending human-likeness evaluation from short-term to long-term behaviors and from individual to population-level patterns. (3) We implement our framework on online shopping, open-topic Q&A, and urban mobility datasets, evaluating mainstream LLMs. Experimental results validate its effectiveness, underscoring the need to enhance AI's human-like capabilities.

## 2. Methodology

### 2.1. Overall Framework

As shown in Figure 1, our framework consists of four key components: behavior dataset, LLM simulator, user encoder, and evaluator.

Publicly available high-quality human behavior data serve as the foundation. We define high-quality data based on three criteria: (1) multiple behaviors per user, (2) diverse

users, and (3) realistic, subjective, self-initiated behaviors. These conditions ensure robust construction of individual personas and human populations, while also reflecting natural behaviors. The LLM simulator acts as a simulation environment, ensuring LLMs accurately replicate sequences of user behaviors. The user encoder learns behavior patterns and encodes them as high-dimensional vectors that capture the underlying behavior distribution. Finally, the evaluator applies multi-dimensional metrics based on rationality, consistency, and diversity to assess human-likeness and believability. The following sections detail each component.

## 2.2. Behavior Dataset

**Amazon Movies&TV** (Hou et al., 2026): Amazon, a major online shopping platform, includes a "Movies& TV" category where users interact with items based on personal interests. In simulating future interactions, an LLM must accurately capture these preferences—failure to do so leads to deviations from real behavior patterns. For each user $u$, there exists an interaction sequence $\mathbf{x}_u = \langle a_1, a_2, ..., a_{|\mathbf{x}_u|} \rangle$, where $a_*$ represents an interacted item in chronological order.

**Stackexchange Worldbuilding**[1]: StackExchange hosts topic-specific Q&A sites, including Worldbuilding, where users discuss imaginary worlds using science, geography, and culture. This domain is highly subjective, with answers shaped by users' knowledge, thinking patterns, and tone. When simulating a user's responses, if the LLM fails to capture these traits, its responses may reveal an AI-generated style, may be completely unrelated to the user's actual replies, or even result in contradictions with the user's other replies. Here, $\mathbf{x}_u$ is unordered and $a_* = [q_*, r_*]$ represents the user's response $r_*$ to a question $q_*$.

**Foursquare Check-ins** (Yang et al., 2015; 2016): Foursquare, a global location-based social platform, records user check-ins at Points of Interest (POIs), reflecting their mobility patterns over time. Accurately simulating future check-ins requires an LLM to understand these spatiotemporal preferences—failure to do so results in deviations from real-world movement patterns. Here $a_*$ represents a visited venue in chronological order. For each recorded check-in, we retain attributes such as category name, visit time, latitude, longitude, city name, and country name, preserving key contextual information for mobility pattern analysis.

More dataset details are in Appendix A.

## 2.3. LLM Simulator

To enable accurate behavior simulation, we design a multi-turn simulation environment where the LLM generates a

sequence of actions for each user. Following (Tseng et al., 2024), we use the mainstream prompt-based simulation method and carefully design the prompt based on three components: (1) Instruction - Directs the LLM to understand user preferences and act as the user to simulate their future behaviors. (2) Personal Profile – Represents user preferences, derived from the first half of their behavior sequence, guiding the LLM to simulate the second half over multiple turns. Each generated action is added to the profile to inform subsequent simulations. (3) Context – Provides scenario-specific information. In online shopping, the context consists of a set of candidate items. The LLM is asked to select the item it believes the user would prefer next. In open-topic Q&A, the context is simply the current question to be answered. In urban mobility, the context consists of candidate venues. The LLM is asked to choose where the user is most likely to visit at a given time. Importantly, each prompt is carefully engineered rather than ad hoc. We iteratively refine its wording and structure through pilot experiments to ensure stable instruction-following behavior across multi-turn simulations for different LLMs. Full prompts are in Appendix D.

## 2.4. User Encoder

To systematically compare LLM-simulated behaviors with real user behaviors, we train two encoders: an item encoder $g()$ and a user encoder $f()$, which encode single behavior and behavior sequence separately into high-dimensional preference vectors. Through training, the encoders learn behavior patterns from real human distributions, enabling robust evaluation of the human-likeness of AI-simulated behaviors. Given a behavior sequence: $\mathbf{x}_u = \langle a_1, a_2, ..., a_{|\mathbf{x}_u|} \rangle$, we first model item embedding $\mathbf{e}_{a_i} = g(a_i)$ and then model user embedding $\mathbf{e}_u = f(\mathbf{e}_{a_1}, \mathbf{e}_{a_2}, ..., \mathbf{e}_{a_{|\mathbf{x}_u|}})$.

In online shopping scenario, we use the classic sequential user model: SASRec (Kang & McAuley, 2018; Lei et al., 2023), where $g()$ is a learned embedding table based on item ID, and $f()$ is a two-layer transformer encoder (Vaswani et al., 2017) with uni-directional attention mask. We use next-item prediction task to optimize our encoders:

$$\mathcal{L} = -\sum_{u \in S_u} \sum_{t \in [1,2,...,|\mathbf{x}_u|-1]} \log \left( \frac{e^{sim(\mathbf{e}_{u_{1:t}}, \mathbf{e}_{a_{t+1}})}}{\sum_{a_k \in S_a} e^{sim(\mathbf{e}_{u_{1:t}}, \mathbf{e}_{a_k})}} \right) \quad (1)$$

Here $S_u$ and $S_a$ are the user set and item set respectively. $sim()$ is cosine similarity and $\mathbf{e}_{u_{1:t}} = f(\mathbf{e}_{a_1}, \mathbf{e}_{a_2}, ..., \mathbf{e}_{a_t})$

In open-topic Q&A scenario, $g()$ is a powerful text embedding model e5-base-v2 (Wang et al., 2022; Lei et al., 2024). We obtain item embeddings by concatenating the question

---

[1]https://archive.org/details/stackexchange

and response. Since questions lack strong temporal relationships, the user encoder $f()$ uses a two-layer transformer encoder with a bidirectional attention mask, using average pooling of the last layer's embeddings as the user representation. For optimization, we push the embedding of each behavior to be close to the embedding of the user's remaining behaviors. This helps the model learn user's unique patterns such as tone and viewpoints in Q&A behaviors. The loss function is:

$$\mathcal{L} = -\sum_{u \in S_u} \sum_{t \in [1,2,...,|\mathbf{x}_u|]} \log \left( \frac{e^{sim(\mathbf{e}_{u_{\neg t}}, \mathbf{e}_{a_t})}}{\sum_{a_k \in N_a} e^{sim(\mathbf{e}_{u_{\neg t}}, \mathbf{e}_{a_k})}} \right) \quad (2)$$

Here $N_a$ are behaviors sampled from other users and $\mathbf{e}_{u_{\neg t}} = f(\mathbf{e}_{a_1}, ..., \mathbf{e}_{a_{t-1}}, \mathbf{e}_{a_{t+1}}, ..., \mathbf{e}_{a_{|\mathbf{x}_u|}})$

In urban mobility scenario, $g()$ is e5-base-v2. We obtain item embeddings by concatenating all attributes of each check-in. User encoder $f()$ is the same as in Q&A scenario but uses a uni-directional attention mask. We use equation 1 for optimization, hoping that the user embedding can be as close as possible to the embedding of next real check-in. In order to train a more expressive model, $S_a$ is a set of hard negatives: we randomly sample locations in the same city as the real check-in.

## 2.5. Evaluator

We propose three principles for evaluating LLM simulated behaviors. At the micro level, we assess rationality and consistency in individual user simulations. At the macro level, we examine the diversity of simulated behaviors across the user population. Real user behavior corresponding to the simulated behavior serves as a benchmark for a relative performance comparison to assess how closely the LLM's simulated behavior resembles human behavior.

### 2.5.1. RATIONALITY

We evaluate if LLM behavior aligns with human expectations. Since it simulates actions based on user profiles, the simulated behavior should reflect the user's personalized preferences and maintain a strong correlation with the personal profile. We divide rationality into two parts:

**Distinguishability**. Inspired by the Turing test, we evaluate whether LLM-generated behaviors are distinguishable from real human behaviors. We construct a dataset with a 1:1 mix of real and simulated sequences, randomly split into training and test sets. A classifier $C_1()$ is trained to predict whether a given sequence $\langle a_1, a_2, ..., a_n \rangle$ is human or LLM-generated:

$$label = C_1(f(\mathbf{e}_{a_1}, \mathbf{e}_{a_2}, ..., \mathbf{e}_{a_n})) \quad (3)$$

We measure LLM performance using the test set *error rate*. A higher error rate indicates that the LLM's behavior is harder to distinguish from real human behavior, demonstrating stronger human-like capabilities. For real user performance, we use a dataset of real behavior sequences but randomly replace half of the labels with 0 (indicating LLM behavior) to introduce noise, testing whether $C_1()$ can still distinguish them.

**Predictability**. When simulating user behavior, LLMs should generate persona-driven responses consistent with the user's profile. Given the behavior sequence: $\mathbf{x}_u = \langle a_1, a_2, ..., a_{|\mathbf{x}_u|} \rangle$, the first half of the sequence $\langle a_1, a_2, ..., a_k \rangle$ serves as the profile guiding the LLM in generating the second half $\langle a_{k+1}^{llm}, a_{k+2}^{llm}, ..., a_{|\mathbf{x}_u|}^{llm} \rangle$, where $k = \lfloor |\mathbf{x}_u|/2 \rfloor$. The *predictability score* is calculated as follows:

$$Score = \frac{1}{|S_u|} \sum_{u \in S_u} \frac{1}{|\mathbf{x}_u| - k} \sum_{t \in [k+1,...,|\mathbf{x}_u|]} sim(\mathbf{e}_*, \mathbf{e}_{a_t^{llm}}) \quad (4)$$

$$\mathbf{e}_* = f(\mathbf{e}_{a_1}, ..., \mathbf{e}_{a_k}, \mathbf{e}_{a_{k+1}^{llm}}, ..., \mathbf{e}_{a_{t-1}^{llm}}) \quad (5)$$

Intuitively, if the LLM's simulated behavior aligns more closely with the user's personality, the user embedding should assign a higher similarity score to the LLM-generated behavior, resulting in a higher predictability score.

### 2.5.2. CONSISTENCY

Consistency evaluates how well LLMs maintain stable long-term behavior patterns. Typical users exhibit steady shopping interests, writing styles, thinking patterns, and moving patterns, while LLMs may show inconsistencies due to uncertain outputs and the potential for omitting earlier information in lengthy contexts, leading to abrupt shifts or contradictions. Formally, we partition the dataset by user into training, validation and test sets. A classifier $C_2()$ is trained to determine whether two behavior segments belong to the same user:

$$label = C_2(f(\mathbf{e}_{a_1}, \mathbf{e}_{a_2}, ..., \mathbf{e}_{a_k}) \oplus f(\mathbf{e}_{a_{k+1}}, \mathbf{e}_{a_{k+2}}, ..., \mathbf{e}_{a_n})) \quad (6)$$

Here $\oplus$ denotes vector concatenation. For positive examples, we randomly split a user's real behavior into two parts, $\mathbf{x}_{u^l} = \langle a_1, a_2, ..., a_k \rangle$ and $\mathbf{x}_{u^r} = \langle a_{k+1}, a_{k+2}, ..., a_n \rangle$. For negative examples, $\mathbf{x}_{u^r}$ is randomly sampled from another user. During evaluation, we split LLM-generated behaviors for each test user into two parts and assess their consistency. The *accuracy score* represents the proportion of simulated users whose two behavior segments are deemed consistent by $C_2()$.

### 2.5.3. DIVERSITY

We evaluates whether the large-scale simulated behaviors generated by LLM align with the behavior distribution of

*Table 1.* Main Experiment on all datasets. Cons, Dist, Pred, Div and Avg refer to Consistency, Distinguishability, Predictability, Diversity, and average score of four metrics. All models we evaluate are instruction-tuned versions. For real user, we calculate metrics using real user behavior instead of LLM simulated behavior. For Llama-3.1-405B, we use fp8 version due to GPU memory limitations. Other models use bf16 inference.

| Dataset | Amazon Movies&TV | | | | | Stackexchange Worldbuilding | | | | | Foursquare Check-ins | | | | |
|---|---|---|---|---|---|---|---|---|---|---|---|---|---|---|---|
| Task | Cons↑ | Dist↑ | Pred↑ | Div↑ | Avg↑ | Cons↑ | Dist↑ | Pred↑ | Div↑ | Avg↑ | Cons↑ | Dist↑ | Pred↑ | Div↑ | Avg↑ |
| Real user | 0.8700 | 0.4820 | 0.2461 | 0.6484 | 0.5616 | 0.8083 | 0.4750 | 0.8222 | 0.8962 | 0.7504 | 0.9150 | 0.4734 | 0.8495 | 0.9372 | 0.7938 |
| Phi-3-mini | 0.3440 | 0.1480 | 0.0102 | 0.5347 | 0.2592 | 0.7410 | 0.0050 | 0.7887 | 0.7416 | 0.5691 | 0.2475 | 0.0641 | 0.8229 | 0.9263 | 0.5152 |
| Phi-3-medium | 0.5440 | 0.1465 | 0.0442 | 0.5385 | 0.3183 | 0.7333 | 0.0200 | 0.7901 | 0.7375 | 0.5702 | 0.3475 | 0.0828 | 0.8248 | 0.9286 | 0.5459 |
| Mistral-small | 0.7000 | 0.1690 | 0.0871 | 0.5649 | 0.3803 | 0.7500 | 0.0050 | 0.7940 | 0.7301 | 0.5698 | 0.1925 | 0.0563 | 0.8194 | 0.9244 | 0.4982 |
| Mistral-large | 0.8480 | 0.1920 | 0.1277 | 0.5864 | 0.4385 | 0.7667 | 0.0750 | 0.7950 | 0.7521 | 0.5972 | 0.5240 | 0.1390 | 0.8320 | 0.9342 | 0.6073 |
| Qwen2.5-3B | 0.2100 | 0.1835 | -0.0012 | 0.5366 | 0.2322 | 0.7167 | 0.0050 | 0.7862 | 0.7030 | 0.5527 | 0.3700 | 0.1078 | 0.8268 | 0.9300 | 0.5587 |
| Qwen2.5-14B | 0.7240 | 0.1835 | 0.1017 | 0.5780 | 0.3968 | 0.7600 | 0.0350 | 0.7987 | 0.6823 | 0.5690 | 0.8475 | 0.2000 | 0.8410 | 0.9343 | 0.7057 |
| Qwen2.5-72B | 0.8160 | 0.2380 | 0.1370 | 0.6092 | 0.4501 | 0.7610 | 0.0900 | **0.8015** | 0.7110 | 0.5909 | **0.8775** | 0.2203 | 0.8424 | 0.9351 | **0.7188** |
| Qwen3-4B | 0.4560 | 0.1290 | 0.0439 | 0.5660 | 0.2987 | 0.7120 | 0.0150 | 0.7880 | 0.7323 | 0.5618 | 0.4800 | 0.1219 | 0.8304 | 0.9305 | 0.5907 |
| Qwen3-14B | 0.6140 | 0.1790 | 0.0907 | 0.5804 | 0.3660 | 0.7450 | 0.0450 | 0.7949 | 0.7415 | 0.5816 | 0.6375 | 0.1859 | 0.8354 | 0.9323 | 0.6477 |
| Qwen3-32B | 0.6720 | 0.1930 | 0.0934 | 0.5936 | 0.3880 | 0.7590 | 0.1000 | 0.7990 | 0.7553 | 0.6033 | 0.7175 | 0.2141 | 0.8379 | 0.9332 | 0.6756 |
| Llama-3.1-8B | 0.5660 | 0.1530 | 0.0482 | 0.5456 | 0.3282 | 0.7000 | 0.0650 | 0.7909 | 0.7445 | 0.5751 | 0.2025 | 0.0672 | 0.8197 | 0.9247 | 0.5035 |
| Llama-3.1-70B | 0.8480 | 0.2140 | 0.1371 | 0.5972 | 0.4491 | 0.7617 | 0.0850 | 0.7997 | 0.7556 | 0.6005 | 0.7350 | 0.1672 | 0.8384 | 0.9342 | 0.6687 |
| Llama-3.1-405B | 0.8540 | 0.2200 | 0.1466 | 0.6035 | 0.4560 | 0.7700 | 0.1150 | 0.7968 | 0.7601 | 0.6105 | 0.8520 | 0.2032 | 0.8410 | 0.9358 | 0.7080 |
| GPT-4o | **0.8800** | **0.3800** | **0.1503** | **0.6106** | **0.5052** | **0.7833** | **0.1520** | 0.7981 | **0.7652** | **0.6247** | 0.8375 | **0.2438** | **0.8435** | **0.9362** | 0.7153 |

real user populations. At a macro level, the simulated user set should reflect diverse real-world behavior patterns rather than collapsing into a narrow range or producing outliers that deviate from actual behavior patterns of users. We divide the dataset by user into a reference group (using real user behaviors) and an experimental group (using LLM-generated behaviors). When the sample size of both groups is sufficiently large, the distribution of user embeddings should be similar between the two groups. For each user in the reference group, we find the nearest neighbor in the experimental group and compute the cosine similarity, plotting the user-similarity curve. The *diversity score* is the average area under the curve. A higher score indicates that each user in the reference group can find a simulated user in the experimental group with more similar preferences. Considering that the real users' behavior preferences in the reference group are evenly distributed, this implies that the user distribution in the experimental group more closely matches the real user distribution in the reference group, and is a more diversified distribution that conforms to real user behavior patterns.

## 2.6. Scalability and Generalizability

We validate our framework using three datasets, but it can be applied to more domains with open-ended behaviors, such as Yelp for offline dining preferences, Twitter for social networking services, Reddit for online forums, IMDb for movie reviews, and Blogger for online blogging. Any domain can be categorized into either candidate-based decision spaces (Amazon and Foursquare) or free-form generative spaces (Worldbuilding), with a consistent implementation, demonstrating the framework's generality and scalability.

## 3. Experiments

### 3.1. Experimental Setup

#### 3.1.1. LLM BACKBONES

We evaluate the following models, all of which are instruction-tuned versions: Llama-3.1 (8B, 70B, and 405B) (Grattafiori et al., 2024); Qwen2.5 (3B, 14B, and 72B) (Qwen et al., 2025); Qwen3 (4B, 14B, and 32B) (Yang et al., 2025); Phi-3 (mini (3.8B) and medium (14B)) (Abdin et al., 2024); Mistral (Mistral-Small-Instruct-2409 (22B) and Mistral-Large-Instruct-2407 (123B)) (Jiang et al., 2023a); GPT-4o (gpt-4o-2024-05-13) (Achiam et al., 2023). Model details are in Appendix B.1 .

#### 3.1.2. IMPLEMENTATION DETAILS

For the LLM simulator, we utilize the vLLM (Kwon et al., 2023) package for efficient inference, setting the temperature to 0.7 and top-p to 0.95. We run all our experiments on 8 NVIDIA H100-80GB GPUs and 4 NVIDIA A100-80GB GPUs. When training the user encoder and classifier, we perform hyperparameter tuning to ensure that the model is fully trained, and the classifier is a 2-layer MLP. More details are in Appendix B.2. Code is available at https://github.com/microsoft/AnthropomorphicIntelligence

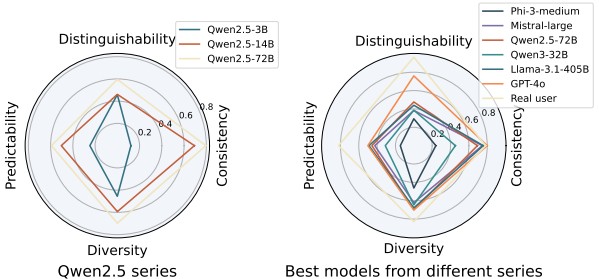

*Figure 2.* Performance comparison of different models on the Movies&TV dataset. We normalize values of each dimension.

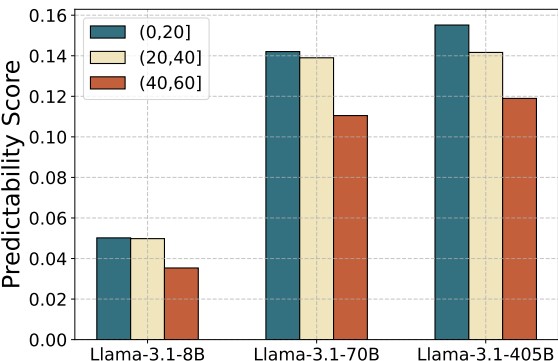

*Figure 3.* Impact of sequence length on predictability on Movies&TV dataset.

## 3.2. Overall Results

The overall results are shown in Table 1. To enhance comparison, Figure 2 presents radar charts for models within the same series and the best models across series on Movies&TV dataset. Additional radar charts for the other two datasets are in Appendix C.1. Key observations:

**Significant gap remains between the best model and humans.** Within the same series, performance improves consistently with model size, aligning with LLM scaling laws (Kaplan et al., 2020). Across different model series, GPT-4o achieves the best overall performance, exhibiting the strongest human-like capabilities, followed by Llama-3.1 and Qwen2.5. However, even the best model lags behind real users by approximately 10%, 17%, and 10% on average on three datasets, respectively, indicating a substantial gap. Small models (<10B) generally perform poorly across all aspects due to their insufficient intelligence to comprehend user preferences and execute simulation tasks. Their generated behavior tends to be more random and illogical, resulting in irrationality, inconsistency, and lacks distribution diversity.

**Different models excel in different aspects.** Performance across metrics varies among open-source models. Llama-3.1 models perform better in consistency and diversity, while Qwen2.5 models lead in distinguishability and predictability. This suggests that Qwen2.5 models better mimic instant human-like behaviors, whereas Llama-3.1 models offer more stable long-term simulations and generate more diverse responses that better align with real user distributions. Our multi-dimensional evaluation framework provides fine-grained insights into these strengths and weaknesses, guiding targeted improvements in LLM training.

**Worldbuilding poses the greatest challenge.** Model performance differs across domains. Among them, LLMs struggle more with the Worldbuilding dataset than the other two, with even the best model trailing real users by 17% on average. This is likely due to the free-form response format, which is inherently more challenging than selecting among a predefined set of candidates in other datasets. Unlike struc-

tured responses, open-topic Q&A requires LLMs not only to replicate user tone but also to grasp users' knowledge scope and reasoning process to generate answers that users themselves would conceive.

In summary, While prior research (Mei et al., 2024; Jiang et al., 2023b) has highlighted GPT-4o's strong human-like abilities, our evaluation shows that even the best models still fall short of real user behavior. This gap underscores the challenge of large-scale, multi-domain behavior simulation, which is more demanding than short, goal-driven conversations or psychological assessments where potential data leakage may occur. Our findings highlight the need for further improvements to bridge the gap between LLM-generated and real user behaviors.

## 3.3. Further Analysis

### 3.3.1. PREDICTABILITY

We analyze the impact of sequence length on the predictability of LLM-simulated behavior by dividing sequences into three length-based buckets. As shown in Figures 3, on Movies&TV dataset, predictability decreases with length. A possible explanation is that with longer simulated sequences, the model's behavior increasingly deviates from the real user's behavior pattern. Because items are discrete and distributed more evenly in vector space, longer simulated sequences introduce more randomness in preferences, reducing predictability. However, in Worldbuilding and Foursquare dataset, predictability increases with length. We put this part of analysis in Appendix C.2.

### 3.3.2. CONSISTENCY

Inspired by (Gupta et al., 2024), we explore LLM consistency in repeated behaviors, examining whether simulated behavior remains stable across semantically identical prompts. On Movie&TV dataset, we let LLM simulate

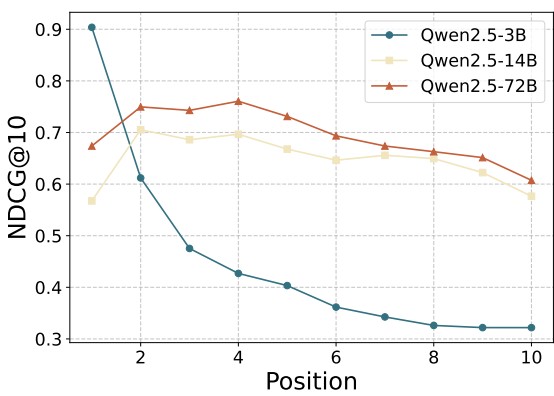

*Figure 4.* The impact of placing target item in different positions of the candidate set on Movies&TV dataset.

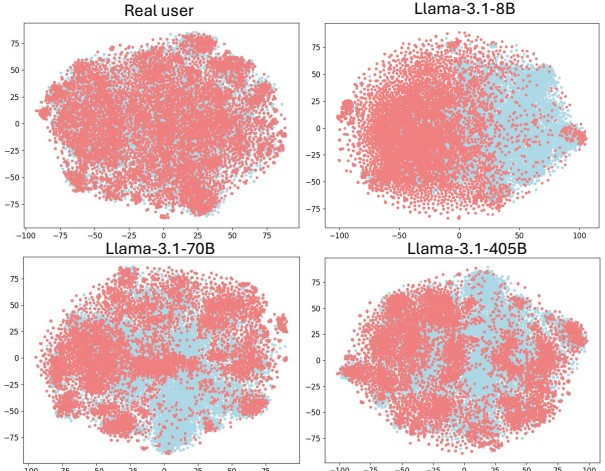

*Figure 5.* t-SNE visualization of user embeddings on Movies&TV dataset. The red points are from experimental group and blue points are from reference group.

behaviors of sorting the candidate set. We get different prompts by varying the target item's position within the candidate set and measure stability using NDCG@10. To ensure stable outputs, we set the temperature to 0.01 and top-p to 1 (full prompt in Appendix D). As shown in Figure 4, larger models exhibit greater stability. Qwen2.5-3B's performance fluctuates significantly with target item position, often ranking earlier-positioned items higher. This suggests smaller models struggle with understanding how to simulate human behavior, making them less robust to prompt variations.

### 3.3.3. DIVERSITY

To illustrate LLM performance in diversity, we apply t-SNE (Van der Maaten & Hinton, 2008) to reduce user embeddings from both the experimental and reference groups to two dimensions and plot them as a scatter plot. As shown

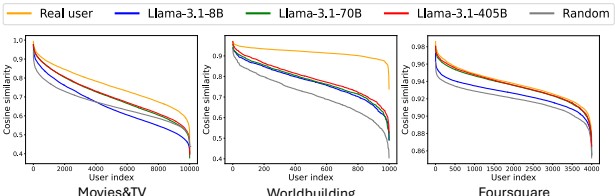

*Figure 6.* User-Similarity curves on three datasets. Similarity scores of users in each figure are arranged in descending order. Full results are in Appendix C.3.

in Figure 5, LLM-simulated behaviors exhibit distribution collapse, which diminishes as model size increases. We also plot user-similarity curves (Figure 6). A smaller area under the curve means real users in the reference group struggle to find similar users in the experimental group, indicating greater deviation from real behavior distribution. On Worldbuilding dataset, a significant gap between real user curves and model curves highlights the challenge of simulating Q&A behavior. In contrast, the Movies&TV and Foursquare dataset, which involves selecting items from a fixed candidate set, has a smaller solution space, making simulation easier.

### 3.4. Ablation Study

#### 3.4.1. IMPACT OF PRE-TRAINED USER ENCODERS

We examine the impact of different user encoders in Q&A domain. For comparison, we train user encoder on another two models: bge-base-en-v1.5 (Xiao et al., 2024) and e5-large-v2. As shown in Table 2, stronger user encoders (with lower validation loss) improve performance across all evaluation metrics. However, the relative performance trends among LLMs and real users remain consistent. Full results are in Appendix C.4.1.

#### 3.4.2. IMPACT OF CLASSIFICATION HEADS

For the classifier, we use by default a well-trained user encoder with a MLP classification head. Since the encoder has effectively captured complex user behavior patterns, a simple classification head suffices. We explore different classification heads on Movies&TV dataset: 1-layer MLP, 2-layer MLP, 3-layer MLP, 2-layer MLP with joint user encoder training, and XGBoost (Chen & Guestrin, 2016) (using fixed user embeddings as features). In Table 3, 2-layer MLP achieves the best performance. Except for the poor 1-layer MLP, all settings, after sufficient training, maintain consistent relative performance trends among LLMs and real users despite absolute differences. Full results are in Appendix C.4.2.

*Table 2.* The best validation loss when training the encoder, and the performance on consistency when using the encoder to evaluate LLMs.

| Model | Val Loss | Real User | Qwen 3B | Qwen 14B | Qwen 72B |
|---|---|---|---|---|---|
| **E5-base** | 0.6237 | 0.8083 | 0.7167 | 0.7600 | 0.7610 |
| **Bge-base** | 0.6639 | 0.8013 | 0.7125 | 0.7530 | 0.7570 |
| **E5-large** | **0.5561** | **0.8110** | **0.7207** | **0.7676** | **0.7695** |

*Table 3.* Performance on consistency when using different classification heads to evaluate LLMs.

| Model | Real User | GPT 4o | Qwen 3B | Qwen 14B | Qwen 72B |
|---|---|---|---|---|---|
| **1-layer MLP** | 0.5600 | 0.4000 | 0.1860 | 0.3820 | 0.4420 |
| **2-layer MLP** | **0.8700** | **0.8800** | **0.2100** | **0.7240** | 0.8160 |
| **3-layer MLP** | 0.8660 | 0.8740 | 0.1940 | 0.6940 | **0.8340** |
| **Joint Training** | 0.8080 | 0.6600 | 0.2060 | 0.5320 | 0.6400 |
| **XGBoost** | 0.8260 | 0.7480 | 0.0840 | 0.5180 | 0.6780 |

## 3.5. Case Study

To better compare simulated behaviors across models, we provide an example on Worldbuilding dataset in Figure 7. This case reveals clear differences in rationality and consistency between the two models. Llama-3.1-405B aligns more closely with the real user by consistently grounding its reasoning in real-world institutional and technical constraints: in the IP lawsuit question it emphasizes jurisdiction, evidence, and enforcement feasibility, and in the NORAD tracking question it continues to treat the problem as a system-level adversarial scenario involving a competent military actor. Although the surface strategy shifts from refuting feasibility to proposing countermeasures, the underlying analytical worldview, tone, and logic remain stable, indicating strong cross-question consistency. However, its rationality largely remains at a bounded and procedural level, prioritizing compliance with known constraints over deeper strategic exploration or novel reasoning like the user, suggesting only a moderate level of overall rationality. In contrast, Phi-3-mini departs significantly from the original author's rational stance, relying on vague narratives such as secret agreements or magical abilities that bypass concrete constraints. Moreover, it exhibits noticeable cross-question drift, portraying Santa first as a legally compliant organizational actor and then as a purely magical figure, resulting in weakened rationality and poor persona consistency across the two simulations. More Examples and a systematic human evaluation are in Appendix C.5 and C.6.

*Figure 7.* Example of a 2-turn simulation on Worldbuilding dataset. Rationality is indicated by text color (green for good, red for poor), while Consistency is denoted by checkmarks and crosses. Full prompt can be found in Appendix D.

## 4. Related Work

**LLM-based Behavior Simulation.** With LLMs achieving human-level intelligence and strong generalization in zero-shot and few-shot tasks, their use as AI agents for behavior simulation is gaining popularity (Gao et al., 2024; Lei et al., 2026). This approach enhances productivity (e.g., reducing annotation costs) and accelerates scientific discovery (e.g., AI-driven feedback collection). Early work by (Park et al., 2023) demonstrated AI-driven social simulation, revealing realistic individual and group behaviors. AI agents have also been explored in gaming (Nagarkar, 2024; Wang et al., 2025b), economic modeling (Sreedhar & Chilton, 2024; Li et al., 2024), and social dynamics across online networks, recommender systems, and urban activities (Gao et al., 2023; Wang et al., 2025a; Xu et al., 2023). Despite these advancements, concerns remain over the believability of AI-generated behaviors (Löhn et al., 2024; Cui et al., 2024; Park et al., 2024). This paper advocates for more rigorous evaluations to address these challenges.

**Evaluations and Human-likeness Test.** Various benchmarks assess LLM capabilities (Chang et al., 2024), including language understanding (Qin et al., 2023), mathematics (Collins et al., 2024), and reasoning (Almeida et al., 2024). This paper focuses on evaluating AI-generated open-ended behaviors for human-likeness and believability, emphasizing distribution-based and sequence-aware evaluations over objective benchmarks for LLMs' humanoid capabilities like theory of mind (Strachan et al., 2024; Wu et al., 2026), emotional intelligence (Sabour et al., 2024; Paech, 2023), or social intelligence (Zhou et al., 2025; Mou et al., 2025). Conversational Turing tests (Jones et al., 2025; Jannai et al., 2023) suggest advanced models like GPT-4 can pass, while (Mei et al., 2024; Hu et al., 2026) assesses AI-human behavioral similarity through classic games and psychological assessments. However, existing tests suffer from data contamination, adversarial biases, and an overemphasis on objective question-answering tasks. We propose a new distribution-based computational framework leveraging large-scale human behavior datasets, eliminating the need for real-time human participants.

## 5. Conclusions

We propose a novel framework that leverages public user behavior data to enable large-scale, distribution-based, and automated evaluation of LLMs' capabilities in simulating open-ended human behaviors. Our evaluation follows three principles—rationality, consistency, and diversity—to ensure believability. Experiments in online shopping, Q&A, and urban mobility scenarios validate our approach, highlighting the need to refine LLMs for greater human-likeness and advancing research in humanoid AI.

## Impact Statement

This work advances research on evaluating the human-likeness of large language models through a computational framework grounded in real-world behavioral data. By studying human behaviors across multiple domains, we contribute to the development of more reliable and human-aligned AI systems. There are many potential societal consequences of our work, none of which we feel must be specifically highlighted here.

We carefully considered ethical and privacy aspects in our study. All datasets used—Amazon Movies & TV, Foursquare Check-ins, and StackExchange Worldbuilding—are publicly available, widely used in prior research, and released in anonymized form. No personally identifiable information is included; user identifiers are anonymized or hashed, and sparsely active users are filtered to further reduce any risk of re-identification. We strictly adhere to dataset terms of use, and do not attempt re-identification or

inference of sensitive attributes. All models used are open-source or accessed via public APIs, and all experiments are conducted solely for scientific research purposes.

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

## A. Details about Dataset

*Table 4.* Statistics of the datasets. For Worldbuilding, #Items refers to the total number of unique questions and answers. #Inters refers to the totol number of interactions. Avg.len refers to the average number of items per user.

| Dataset | #Users | #Items | #Inters | Avg.len |
|---|---|---|---|---|
| Movies&TV | 34,803 | 25,831 | 899,319 | 25.8 |
| Worldbuilding | 2,056 | 147,391 | 111,690 | 54.3 |
| Foursquare | 113,335 | 1,610,221 | 7,876,782 | 69.5 |

In online shopping scenario, we filter out low-frequency items (appearing <6 times) to avoid unrecognized data by LLMs and retain only users with at least 12 interactions. This ensures the LLM has sufficient context to learn preferences and the user encoder can reliably model behavior patterns. In open-topic Q&A scenario, we filter out low-quality answers (score <3) and retain users who have answered at least 7 questions. In urban mobility scenario, we filter out users with fewer than 20 check-ins.

We list the statistics of datasets in Table 4. The statistics of the datasets corresponding to the user encoder and evaluator are shown in Table 5.

*Table 5.* Dataset statistics for user encoder and evaluator. Cons, Dist, Pred, Div refer to Consistency, Distinguishability, Predictability, and Diversity.

| | | Train | Valid | Test |
|---|---|---|---|---|
| **Movies&TV** | Encoder | 795,355 | 43,225 | – |
| | Cons | 42,476 | 2,308 | 500 |
| | Dist | 18,000 | – | 4,000 |
| | Pred | – | – | 10,000 |
| | Div | – | – | 10,000 |
| **Worldbuilding** | Encoder | 101,120 | 4,736 | – |
| | Cons | 3,340 | 371 | 200 |
| | Dist | 1,840 | – | 160 |
| | Pred | – | – | 1,000 |
| | Div | – | – | 1,000 |
| **Foursquare** | Encoder | 93,056 | 10,368 | – |
| | Cons | 20,000 | 2,000 | 400 |
| | Dist | 7,360 | – | 640 |
| | Pred | – | – | 4,000 |
| | Div | – | – | 4,000 |

## B. Experimental Setup and Hyperparameters

### B.1. LLM Backbones

- **Llama-3.1 (Grattafiori et al., 2024)**: Llama-3.1 is Meta's advanced iteration of the LLaMa model (Touvron et al., 2023). The instruction-tuned version of the pure text model is optimized for multilingual dialogue use cases and outperforms many available open-source and closed-chat models on common industry benchmarks. We utilize three different sizes of the

instruction-tuned model: 8B, 70B, and 405B.

- **Qwen2.5 (Qwen et al., 2025)**: Qwen2.5 is an earlier version of the Qwen large language model series developed by Alibaba. Compared to Qwen2 (Yang et al., 2024), it has greatly improved its capabilities in following instructions, generating long texts, and producing structured outputs. We utilize three sizes of the instruction-tuned versions: 3B, 14B, and 72B.

- **Qwen3 (Yang et al., 2025)**: Qwen3 is the latest generation of large language models in Qwen series, providing a comprehensive suite of dense models. It delivers groundbreaking advancements in reasoning, instruction-following, agent capabilities, and multilingual support.

- **Phi-3 (Abdin et al., 2024)**: Phi-3 is an open-source AI model series developed by Microsoft. The Phi-3 model is a high-performing and cost-effective small language model (SLM), claiming to outperform models of the same size and those one size larger. We use its mini (3.8B) and medium (14B) sizes of 128k instruction-tuned versions.

- **Mistral (Jiang et al., 2023a)**: The Mistral AI team has released a series of powerful large language models. We utilize two of their instruction-tuned models: Mistral-Small-Instruct-2409 (22B) and Mistral-Large-Instruct-2407 (123B).

- **GPT-4o (Achiam et al., 2023)**: GPT-4o is one of OpenAI's advanced multimodal model. This model features a context length of 128K and has a October 2023 knowledge cutoff. The specific version we use is gpt-4o-2024-05-13.

### B.2. Implementation Details

We simulate behaviors of 10,000/1,000/4,000 users in the online shopping, open-topic Q&A and urban mobility scenarios, respectively. For cost considerations, we limit the simulation with GPT-4o to 500/200/400 users. In the online shopping scenario, the candidate item set consists of the user's next favorite item and 99 randomly sampled negative items. In the urban mobility scenario, the candidates consist of the user's next check-in and 9 randomly sampled locations in the same city. When encoding user embeddings, we truncate overly long sequences, with a maximum sequence length of 60/30/60 for the online shopping, open-topic Q&A and urban mobility scenarios.

When training the user encoder, we fine-tuned learning rate in {1e-3, 3e-4, 1e-4, 3e-5, 1e-5, 3e-6} as well as other possible hyperparameters. For online shopping scenario: learning rate = 1e-3, embedding size = 128, hidden dropout probability = 0.4, attention dropout probability = 0.1, max

sequence length = 60. For Q&A scenario: learning rate = 3e-5, max sequence length = 30, item encoder is not frozen. For urban mobility scenario: learning rate = 1e-4, max sequence length = 60, item encoder is not frozen.

When training classification heads, we fine-tuned learning rate in {1e-3, 8e-4, 5e-4, 1e-4, 8e-5, 5e-5, 1e-5, 8e-6}. For online shopping scenario: learning rate = 8e-4 / 5e-4 for Consistency / Distinguishability. For Q&A scenario: learning rate = 1e-4 / 1e-4 for Consistency / Distinguishability. For urban mobility scenario: learning rate = 1e-4 / 1e-4 for Consistency / Distinguishability.

## C. More Experimental Results

### C.1. Overall Results

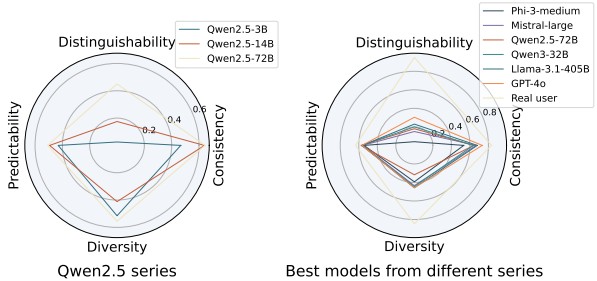

*Figure 8.* Performance comparison of different models on Worldbuilding dataset. We normalize values of each dimension.

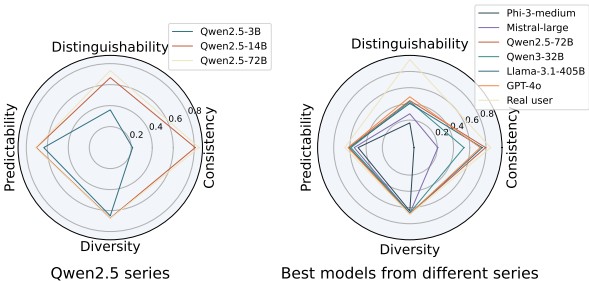

*Figure 9.* Performance comparison of different models on Foursquare dataset. We normalize values of each dimension.

Figure 8 and 9 are performance radar charts on Worldbuilding and Foursquare dataset.

In the initial experiments on the Worldbuilding dataset, we observe that responses generated by LLMs tend to exhibit a noticeable "AI tone", especially for smaller models. These responses are often highly structured and meticulously enumerate key points, which differs from the personalized behaviors we aim to evaluate. In open-ended writing settings, both surface-level style and deeper behavioral patterns are important components of human behavior. Therefore, for Rationality and Diversity, our encoder is trained on human

behaviors containing both aspects, and we do not expect the framework itself to be invariant to tone or writing style. However, for Consistency, we observe an interaction specific to current LLM limitations: although simulated responses may be inconsistent at the level of deeper viewpoints, they often remain highly consistent in AI-style tone and formatting. Since the original encoder jointly considers both aspects, this stylistic consistency can become a shortcut and artificially inflate consistency scores. We emphasize that this is not a limitation of the framework itself, but rather an artifact introduced by the current generation characteristics of LLMs.

To mitigate this issue, we use Llama3.1-70B to rewrite both real-user and LLM-generated responses while preserving their underlying viewpoints and behavioral content. This post-processing removes surface-level language style and formatting effects, allowing Consistency to focus on deeper behavioral patterns and provide a more meaningful estimate of current LLM simulation ability in open-ended writing scenarios. Notably, this issue does not arise in the other two domains.

To validate that rewriting does not introduce substantial semantic distortion, we conduct a human evaluation on 300 rewritten samples (20 randomly selected cases per model in Table 1). A graduate student rates content fidelity using a 1–5 Likert scale (1: content missing; 2: largely missing; 3: moderately missing; 4: minor information loss; 5: fully preserved). The average score is 4.96, with only 8 samples receiving a score of 4 and 2 samples receiving a score of 3, indicating that rewriting preserves semantic content well and introduces minimal evaluation bias. The results shown in Table 1 reflect this adjustment. As observed, after removing the influence of language style and text formatting, the consistency scores of smaller models drop substantially, which aligns with our expectations. The rewriting prompt is provided in Appendix D.

## C.2. Predictability Analysis

As shown in Figures 10 and 11, in contrast to the results on Movies&TV dataset, in Worldbuilding and Foursquare dataset, predictability increases with length. In Worldbuilding, there is token overlap between responses, causing embeddings to be closer in the vector space. Additionally, LLM-generated responses tend to carry a certain "AI style". As the length of the simulated sequence increases, answers become more aligned with the AI's style, bringing the vectors closer together and leading to increased predictability. Similarly, in Foursquare, the LLM movement pattern will gradually accumulate and bring the vectors closer.

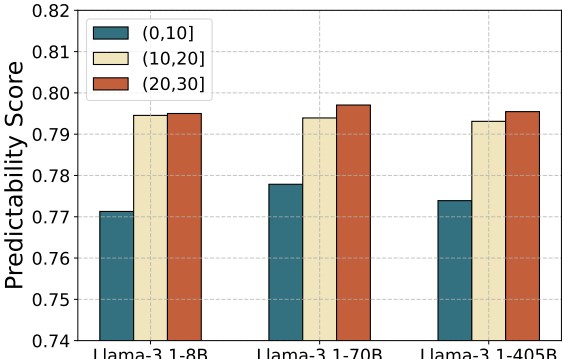

*Figure 10.* Impact of sequence length on predictability on Worldbuilding dataset.

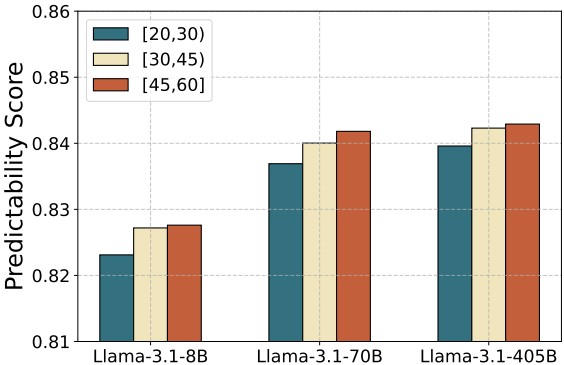

*Figure 11.* Impact of sequence length on predictability on Foursquare dataset.

## C.3. Full User-Similarity Curves

We show the user-similarity curves of all models on all datasets in Figure 12. It can be clearly seen that the areas under the curves of different models are different, which represents the similarity between their simulated distribution and the real distribution of the reference group.

## C.4. Ablation Study

### C.4.1. IMPACT OF PRE-TRAINED USER ENCODERS

Full results of the impact of different pre-trained user encoders are in Table 6 and 7.

### C.4.2. IMPACT OF CLASSIFICATION HEADS

Full results of the impact of different classification heads are in Table 8 and 9.

*Table 6.* LLM simulation performance on Worldbuiding dataset when using bge-base-en-v1.5 for user encoder. Cons, Dist, Pred, Div refer to Consistency, Distinguishability, Predictability, and Diversity.

| Task | Cons↑ | Dist↑ | Pred↑ | Div↑ |
|---|---|---|---|---|
| Real User | 0.8013 | 0.4850 | 0.8003 | 0.8916 |
| Phi-3-mini | 0.7380 | 0.0150 | 0.7700 | 0.7363 |
| Phi-3-medium | 0.7303 | 0.0400 | 0.7736 | 0.7337 |
| Mistral-small | 0.7460 | 0.0250 | 0.7762 | 0.7274 |
| Mistral-large | 0.7602 | 0.0950 | 0.7784 | 0.7477 |
| Qwen2.5-3B | 0.7125 | 0.0100 | 0.7696 | 0.6980 |
| Qwen2.5-14B | 0.7530 | 0.0550 | 0.7794 | 0.6805 |
| Qwen2.5-72B | 0.7570 | 0.1050 | **0.7830** | 0.7102 |
| Qwen3-4B | 0.7090 | 0.0250 | 0.7712 | 0.7293 |
| Qwen3-14B | 0.7411 | 0.0600 | 0.7785 | 0.7385 |
| Qwen3-32B | 0.7545 | 0.1100 | 0.7814 | 0.7505 |
| Llama3.1-8B | 0.6950 | 0.0800 | 0.7725 | 0.7429 |
| Llama3.1-70B | 0.7580 | 0.1000 | 0.7777 | 0.7557 |
| Llama3.1-405B | 0.7640 | 0.1350 | 0.7758 | 0.7583 |
| GPT-4o | **0.7785** | **0.1750** | 0.7820 | **0.7600** |

*Table 8.* Impact of the classification head on consistency metric on Movies&TV dataset.

| Model | 1-layer | 2-layer | 3-layer | Joint Train | XGBoost |
|---|---|---|---|---|---|
| Real User | 0.5600 | 0.8700 | 0.8660 | 0.8080 | 0.8260 |
| Phi-3-mini | 0.2820 | 0.3440 | 0.3280 | 0.3560 | 0.1320 |
| Phi-3-medium | 0.3540 | 0.5440 | 0.5160 | 0.4360 | 0.2820 |
| Mistral-small | 0.3860 | 0.7000 | 0.6840 | 0.5360 | 0.4860 |
| Mistral-large | 0.4220 | 0.8480 | 0.8500 | 0.5920 | 0.6840 |
| Qwen2.5-3B | 0.1860 | 0.2100 | 0.1940 | 0.2060 | 0.0840 |
| Qwen2.5-14B | 0.3820 | 0.7240 | 0.6940 | 0.5320 | 0.5180 |
| Qwen2.5-72B | **0.4420** | 0.8160 | 0.8340 | 0.6400 | 0.6780 |
| Qwen3-4B | 0.3020 | 0.4560 | 0.4350 | 0.4425 | 0.2080 |
| Qwen3-14B | 0.3460 | 0.6164 | 0.6020 | 0.5104 | 0.4895 |
| Qwen3-32B | 0.3580 | 0.6720 | 0.6840 | 0.5340 | 0.5750 |
| Llama3.1-8B | 0.3500 | 0.5660 | 0.5220 | 0.5100 | 0.3220 |
| Llama3.1-70B | 0.4340 | 0.8480 | 0.8420 | **0.6900** | 0.7360 |
| Llama3.1-405B | 0.3740 | 0.8540 | 0.8540 | 0.6780 | **0.7560** |
| GPT-4o | 0.4000 | **0.8800** | **0.8740** | 0.6600 | 0.7480 |

*Table 7.* LLM simulation performance on Worldbuilding dataset when using e5-large-v2 for user encoder. Cons, Dist, Pred, Div refer to Consistency, Distinguishability, Predictability, and Diversity.

| Task | Cons↑ | Dist↑ | Pred↑ | Div↑ |
|---|---|---|---|---|
| Real User | 0.8110 | 0.4700 | 0.8508 | 0.9110 |
| Phi-3-mini | 0.7433 | 0.0050 | 0.8147 | 0.7522 |
| Phi-3-medium | 0.7390 | 0.0200 | 0.8188 | 0.7500 |
| Mistral-small | 0.7545 | 0.0050 | 0.8221 | 0.7426 |
| Mistral-large | 0.7698 | 0.0700 | 0.8235 | 0.7664 |
| Qwen2.5-3B | 0.7207 | 0.0050 | 0.8168 | 0.7203 |
| Qwen2.5-14B | 0.7676 | 0.0300 | 0.8262 | 0.7057 |
| Qwen2.5-72B | 0.7695 | 0.0800 | **0.8298** | 0.7333 |
| Qwen3-4B | 0.7200 | 0.0100 | 0.8080 | 0.7529 |
| Qwen3-14B | 0.7528 | 0.0350 | 0.8152 | 0.7612 |
| Qwen3-32B | 0.7671 | 0.0950 | 0.8193 | 0.7685 |
| Llama3.1-8B | 0.7070 | 0.0600 | 0.8191 | 0.7598 |
| Llama3.1-70B | 0.7697 | 0.0750 | 0.8277 | 0.7653 |
| Llama3.1-405B | 0.7790 | 0.1050 | 0.8263 | 0.7692 |
| GPT-4o | **0.7875** | **0.1450** | 0.8270 | **0.7720** |

*Table 9.* Impact of the classification head on distinguishability metric on Movies&TV dataset.

| Model | 1-layer | 2-layer | 3-layer | Joint Train | XGBoost |
|---|---|---|---|---|---|
| Real User | 0.4835 | 0.4820 | 0.4775 | 0.4760 | 0.4820 |
| Phi-3-mini | 0.2245 | 0.1480 | 0.1435 | 0.1335 | 0.1440 |
| Phi-3-medium | 0.2060 | 0.1465 | 0.1420 | 0.1305 | 0.1400 |
| Mistral-small | 0.2230 | 0.1690 | 0.1590 | 0.1570 | 0.1620 |
| Mistral-large | 0.2270 | 0.1920 | 0.1845 | 0.1740 | 0.1905 |
| Qwen2.5-3B | 0.2615 | 0.1835 | 0.1725 | 0.1670 | 0.1835 |
| Qwen2.5-14B | 0.2285 | 0.1835 | 0.1770 | 0.1715 | 0.1845 |
| Qwen2.5-72B | 0.3080 | 0.2380 | 0.2275 | 0.2040 | 0.2280 |
| Qwen3-4B | 0.1820 | 0.1290 | 0.1180 | 0.1055 | 0.1280 |
| Qwen3-14B | 0.2360 | 0.1790 | 0.1700 | 0.1620 | 0.1785 |
| Qwen3-32B | 0.2650 | 0.1930 | 0.1855 | 0.1705 | 0.1900 |
| Llama3.1-8B | 0.1940 | 0.1530 | 0.1425 | 0.1375 | 0.1580 |
| Llama3.1-70B | 0.2480 | 0.2140 | 0.2090 | 0.1965 | 0.2115 |
| Llama3.1-405B | 0.2745 | 0.2200 | 0.2105 | 0.2005 | 0.2170 |
| GPT-4o | **0.4400** | **0.3800** | **0.4000** | **0.3720** | **0.3810** |

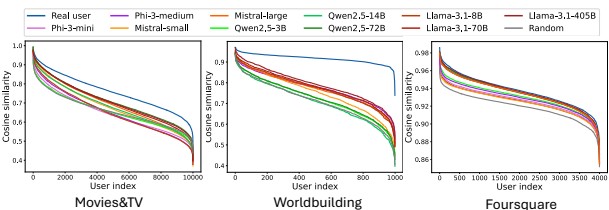

*Figure 12.* Full User-Similarity curves on three datasets. Similarity scores of users in each figure are arranged in descending order.

## C.5. Case Study

In Figure 13, we show a 4-turn simulation on the Movies&TV dataset. The user's purchase history is dominated by romantic dramas (e.g., Fairfield Road, How to Fall in Love) and light comedies (e.g., Moonlight & Mistletoe), with subsequent real purchases (Scents and Sensibility, Priceless) consistently following this preference pattern. Llama-3.1-405B demonstrates strong rationality by correctly inferring the user's genre preferences and selecting semantically aligned items, including comedies (It's a Mad Mad Mad Mad World) and romance-inflected narratives (Fallen Angel), while also maintaining consistency across multiple simulations. In contrast, Phi-3-mini fails to capture the user's underlying preferences, selecting action and horror films (Blade Runner, Halloween Kills) as well as an unrelated music video (The Piano Guys: Live at Red Butte Garden). These predictions are neither genre-consistent with the user's historical purchases nor coherent across the simulation sequence, reflecting deficiencies in both rationality and consistency.

Similarly, in Figure 14, we show a 3-turn simulation on the Foursquare dataset. In the first round of behavior simulation, the user's historical check-in records indicate a clear weekday pattern of visiting work-related locations between 11:00 and 13:00. Llama-3.1-405B correctly captures this pattern, whereas Phi-3-mini fails to do so and instead predicts a café visit, reflecting weaker rationality. In the third round, Phi-3-mini further exhibits a consistency violation. Although the predicted office visit is reasonable in isolation, the time gap between the second and third check-ins is only 13 seconds, while the spatial distance exceeds 20 km, resulting in an implausible transition. This demonstrates its inability to maintain coherent spatiotemporal continuity across consecutive behaviors.

In Figure 15, we aim to illustrate the lack of diversity exhibited by different models when simulating the behaviors of distinct users. We focus on questions that have been answered by multiple users, which allows for a direct comparison of how LLMs differentiate (or fail to differentiate) between users in their simulated responses. For the same speculative question, we observe a clear increase in inter-user response diversity from LLM-simulated users to real

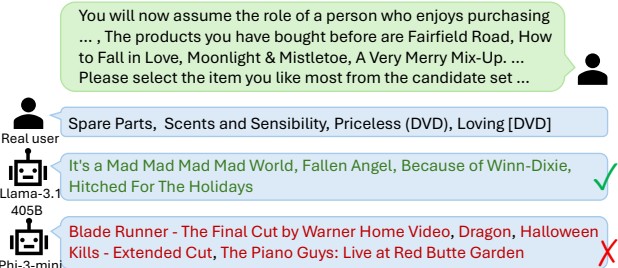

*Figure 13.* Example of a 4-turn simulation on Movies&TV dataset. Rationality is indicated by text color (green for good, red for poor), while Consistency is denoted by checkmarks and crosses. To save space, the prompt in the figure is simplified and the full one can be found in Appendix D.

human users. The responses generated by phi-3-mini resemble each other closely, with different "users" converging on similar explanations centered on canonical genetic mechanisms, indicating limited behavioral differentiation. LLaMA-3.1-405B produces more varied simulated users, whose answers diverge across multiple biological and technological pathways (e.g., genetic, epigenetic, hormonal), reflecting improved but still structured diversity. In contrast, real users exhibit the strongest diversity between individuals: one user frames the problem through an immunological lens, while another constructs a detailed gene-drive–based engineering solution. These responses differ not only in content but also in reasoning style and level of technical depth, demonstrating substantially richer user-level heterogeneity than that observed in model-simulated users.

## C.6. Human Evaluation

To further verify that the conclusions drawn from our computational framework are reliable rather than incidental, we conduct a systematic human evaluation beyond the case studies. We recruit two graduate students specializing in societal AI, neither of whom is involved as a co-author of this work, to assess the generated behaviors. Annotators are blinded to the source of each sequence (i.e., model-generated or real human behavior) as well as to the model identity and size. Given the substantial annotation cost, we restrict the evaluation to the LLaMA family of models and real human behaviors.

Specifically, for each dataset, we randomly sample 100 users from the test set, resulting in 400 evaluation instances in total, including 100 simulated behavior sequences from each model and 100 real user sequences. For Rationality, annotators are provided with a user's personal profile and asked to judge whether the corresponding behavior sequence is consistent with the profile and plausibly originates from the same individual. The same profiles are used for evaluating both real and simulated sequences. For Con-

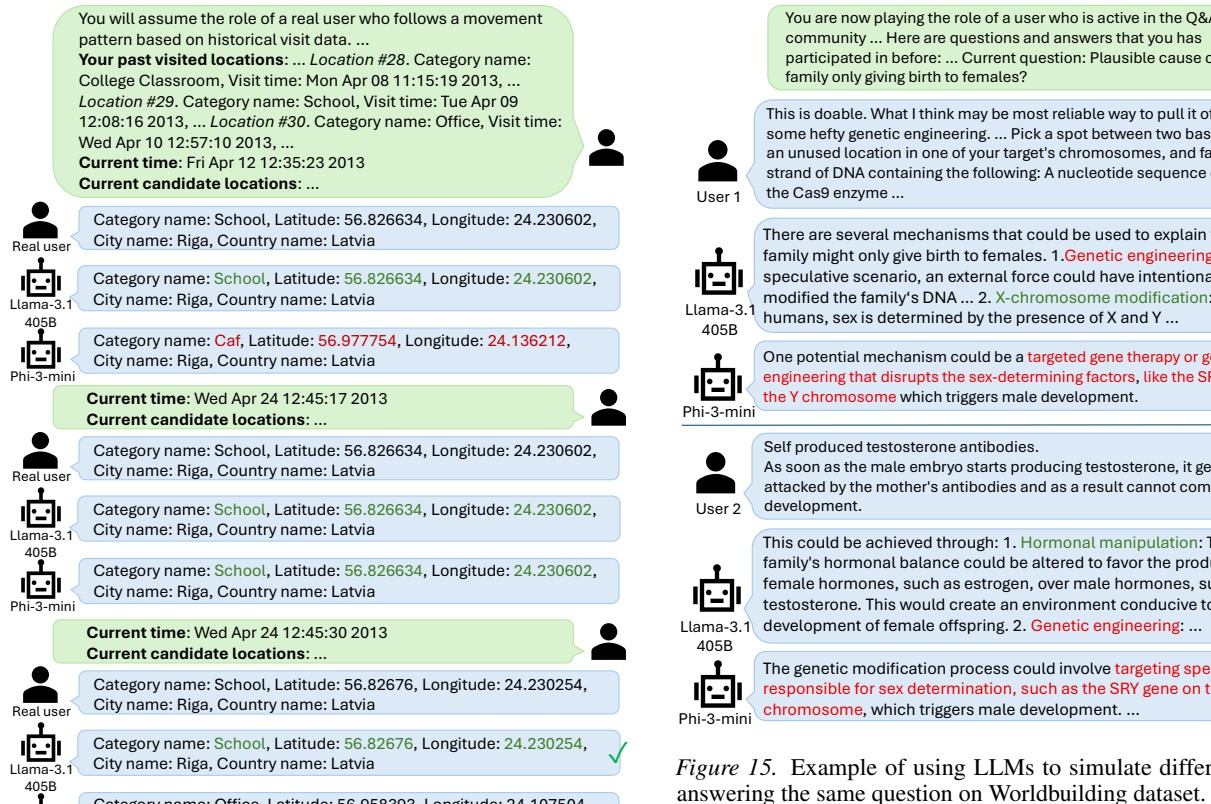

*Figure 14.* Example of a 3-turn simulation on Foursquare dataset. Rationality is indicated by text color (green for good, red for poor), while Consistency is denoted by checkmarks and crosses. Full prompt can be found in Appendix D.

*Figure 15.* Example of using LLMs to simulate different users answering the same question on Worldbuilding dataset. Diversity is indicated by text color (green for good, red for poor).

*Table 10.* Spearman correlation between human ratings and framework scores across datasets ($p < 0.001$).

|  | Movies&TV | Worldbuilding | Foursquare |
|---|---|---|---|
| Rationality | 0.58 | 0.55 | 0.55 |
| Consistency | 0.52 | 0.59 | 0.61 |

sistency, annotators assess whether a behavior sequence exhibits stable and coherent patterns across multiple actions, or whether it contains anomalies such as abrupt shifts or inconsistent behavioral tendencies. For Diversity of Worldbuilding dataset, we first select 100 questions, each originally answered by at least two different users in the dataset. For each question, we retain responses from two distinct users, and correspondingly, models generate two responses by simulating these two users. Annotators then evaluate, at the question level, whether the pair of responses attributed to different users exhibits sufficient diversity. Since Amazon and Foursquare dataset consist of user-specific behavior sequences rather than multiple users responding to a shared prompt, we adopt a group-level evaluation protocol. For each dataset, we randomly sample 150 users and partition them into groups of 10. Annotators then assess whether the sequences within each group exhibit diverse, distinctive, and non-template-like behavioral patterns, rather than collapsing toward homogeneous or average-user-like behaviors. All evaluations are conducted using a 1–5 Likert scale, where 5 indicates the highest degree of human-likeness. The averaged human evaluation scores are reported

in Table 11, 12, and 13. We observe that the overall performance trends are highly consistent with those in Table 1: performance improves steadily with increasing model size across all metrics, yet remains substantially below that of real human behavior. To assess the reliability of human judgments, we compute ordinal Krippendorff's $\alpha$ to measure inter-annotator agreement on the Likert-scale ratings. We obtain agreement scores of 0.75/0.69/0.76 for Rationality/Consistency/Diversity on Movies&TV dataset, 0.75/0.72/0.81 on Worldbuilding dataset, and 0.72/0.79/0.82 on Foursquare dataset, respectively. These results indicate substantial inter-annotator agreement across all datasets and support the robustness of our human evaluation.

We further investigate whether our framework scores align with human judgments at the instance level. Using the same evaluated sequences, we compute the Spearman correlation between human ratings and the framework's continuous logits for Rationality and Consistency (Diversity is excluded

since it is not defined at the instance level). We observe significant positive correlations in Table 10, indicating that sequences receiving higher framework scores are also perceived by human annotators as more human-like.

Overall, these results provide cross-domain external validation, suggesting that our metrics capture meaningful human-likeness rather than merely encoder-dependent similarity, and further demonstrate the generalizability of the framework.

*Table 11.* Human evaluation results on Movies&TV dataset.

|  | Real User | Llama-3.1-8B | Llama-3.1-70B | Llama-3.1-405B |
|---|---|---|---|---|
| Rationality | 4.20 | 1.12 | 2.35 | 2.76 |
| Consistency | 4.42 | 3.10 | 3.94 | 4.11 |
| Diversity | 4.45 | 2.96 | 3.40 | 3.72 |

*Table 12.* Human evaluation results on Worldbuilding dataset.

|  | Real User | Llama-3.1-8B | Llama-3.1-70B | Llama-3.1-405B |
|---|---|---|---|---|
| Rationality | 4.45 | 1.31 | 2.10 | 2.42 |
| Consistency | 4.23 | 2.95 | 3.56 | 3.78 |
| Diversity | 4.52 | 2.35 | 2.68 | 3.10 |

*Table 13.* Human evaluation results on Foursquare dataset.

|  | Real User | Llama-3.1-8B | Llama-3.1-70B | Llama-3.1-405B |
|---|---|---|---|---|
| Rationality | 4.52 | 2.10 | 2.96 | 3.22 |
| Consistency | 4.36 | 1.22 | 3.05 | 3.96 |
| Diversity | 4.28 | 3.14 | 3.56 | 3.88 |

## D. Prompts for LLM Simulator

Figure 16, 17, 18, 19, and 20 are prompt examples used by LLM Simulator on three datasets, where fonts highlighted in blue represent personal profile, fonts highlighted in red represent context and others represent instruction.

---

**Online Shopping Simulator Prompt**

---

You will now assume the role of a person who enjoys purchasing Movies and TV category products on an e-commerce platform.

Given a list of candidate products, your task is to select one product that aligns with your purchasing preferences. Your preferences are represented by the products you have bought in the past. When choosing a product from the candidate products, you should first analyze your past purchases to get your own purchasing preferences, and then follow your preferences to select something your character would likely prefer.

The products you have bought before are: ["Selena [DVD]", "Selena Live - The Last Concert", ...].

The current list of candidate products is: ["Cujo [DVD]", "The Elephant Man", ...].

Your response should include only the name of the product you would buy. Do not include any additional text in your response. Do not output product that is in your past purchases.

*Figure 16.* Online Shopping Simulator Prompt.

---

**Open-topic Q&A Simulator Prompt**

---

You are now playing the role of a user who is active in the Q&A community and likes to answer questions about the worldbuiling domain.

Given the current question, your task is to answer the question as closely as possible to the speaking style, thinking patterns, and opinions of the user you are playing. I will give you some questions that the user has participated in in the past and the user's answers. Your answer should try not to contradict the previous answers, but try to make your answer sounds like the same person answering the question, that is, the speaking style should be consistent, the thinking mode of answering the question should be consistent, and the opinions should also be consistent.

Here are some questions and answers that the user has participated in before:
Question#1: What motivation would people have for mining planetary rings? Answer#1: I don't have a great grasp ...
Question#2: ... Answer#2: ...
...

Now, here is the current question: How can asymmetrically powerful states be balanced?

Please give your answer.

*Figure 17.* Open-topic Q&A Simulator Prompt.

---

**Online Shopping Ranking Simulator Prompt**

---

You will now assume the role of a person who enjoys purchasing Movies and TV category products on an e-commerce platform.

Given a list of candidate products, your task is to sort the products in the candidate list from largest to smallest according to your purchasing preference. Your preferences are represented by the products you have bought in the past. When sorting products in the candidate products, you should first analyze your past purchases to get your own purchasing preferences, and then follow your preferences to sort products by your preferences, i.e. the product you are most likely to like should be ranked first, and the product you dislike the most should be ranked last.

The products you have bought before are: ["Selena [DVD]", "Selena Live - The Last Concert", ...].

The current list of candidate products is: ["Cujo [DVD]", "The Elephant Man", ...].

Your response should include only the names of the products separated by @@. Do not include any additional text in your response. Do not output product that is in your past purchases.

*Figure 18.* Online Shopping Ranking Simulator Prompt.

---

**Urban Mobility Simulator Prompt**

---

Task Description:
You will assume the role of a real user who follows a movement pattern based on historical visit data. Given a time range and a list of candidate locations, your task is to predict the most likely place the user would visit based on past movement behavior.

Input Information:
You will be provided with:
1. A list of past visited locations, including Category name, Visit time, Latitude, Longitude, City name, Country name.
2. Current time range.
3. A list of candidate locations with the same attributes.

Task Requirements:
1. Temporal Analysis : Analyze past visits at similar times of the day to determine the most probable types of places visited during the given time range. Identify if the user has recurring visit patterns within specific timeframes and prioritize locations accordingly.
2. Category Preference Matching: Determine which categories the user frequently visits and prioritize candidates that belong to these high-frequency categories. If the user has an established pattern of visiting a certain category in a specific sequence, favor candidates that align with this sequence.
3. Sequential Pattern Recognition: If the user exhibits a pattern of visiting specific types of places in a recurring sequence, prioritize candidates that fit into these sequences. If the user has a history of linking certain place types (e.g., sports venues followed by sporting goods stores), prioritize candidates that maintain this behavioral flow.
4. Contextual Adaptation: If the user is currently in transit or has recently moved between cities, prioritize locations that match typical post-travel visit behavior. If the user has previously visited an event-based location (e.g., a concert or stadium), consider locations related to follow-up activities such as dining or shopping.
5. Logical Feasibility (Distance & Time Constraints): Assess whether the transition from the user's most recent visit to a candidate location is physically feasible based on distance and time constraints. If a candidate location is too far from the last visited location given the available time, deprioritize it. If a candidate location is unreachable under realistic travel conditions (e.g., due to city infrastructure, transportation limits), it should not be considered a valid choice.
6. Avoid Overgeneralization: Do not assume general behavior such as "users always visit a restaurant at mealtime" unless explicitly supported by past visit patterns. Avoid assigning high priority to categories that the user has never or rarely visited in similar contexts.
Using these rules, select the most likely location from the candidate list.

Output Requirements:
Your response should only include the index number of the selected location from the candidate list (from 1 to {num}). You must not provide any additional text, explanations, or reasoning.

Your past visited locations:
Location #1. Category name: Campground, Visit time: Sat Jul 14 07:47:12 2012, Latitude: 3.173162, Longitude: 101.738167, City name: Kuala Lumpur, Country name: Malaysia
Location #2. Category name: Lake, Visit time: Sun Jul 15 17:52:41 2012, Latitude: 3.17741, Longitude: 101.707027, City name: Kuala Lumpur, Country name: Malaysia
...

Current time range: Fri Jan 04 11:31:15 2013

Current candidate locations:
1. Category name: Mexican Restaurant, Latitude: 3.09198, Longitude: 101.543808, City name: Shah Alam, Country name: Malaysia
2. Category name: Restaurant, Latitude: 2.989335, Longitude: 101.635291, City name: Shah Alam, Country name: Malaysia
...

*Figure 19.* Urban Mobility Simulator Prompt.

---

**Open-topic Q&A Rewriting Prompt**

---

You are an AI assistant tasked with harmonizing both the language style and structure of various responses to the same question. Some responses are written in a continuous style, while others use bullet points or focus on one key point. Please rephrase and unify the responses into a continuous, well-organized paragraph, clearly separating different ideas with proper transitions and ensuring a consistent, neutral and objective tone. Retain the original reasoning and key content of each response. Here is the original response: {origin}. Please rephrase and format this response into coherent and well-structured paragraphs.

*Figure 20.* Open-topic Q&A Rewriting Prompt.

