# OpenReview forum: "A Computational Framework for Evaluating Human-likeness in LLMs' Open-ended Human Behaviors"
_ICML.cc/2026/Conference — ICML 2026 regular_

### Official Review · Reviewer_DTjL · 2026-03-08

**Soundness:** 2
**Presentation:** 3
**Significance:** 2
**Originality:** 2
**Overall Recommendation:** 4
**Confidence:** 4

**Summary:**

This paper proposes a distribution-based framework for evaluating how human-like LLM-generated behaviors are, using three real-world datasets (Amazon shopping, StackExchange Q&A, Foursquare mobility). They train user encoders to embed behavior sequences, then evaluate LLMs across four metrics: consistency, distinguishability, predictability, and diversity. Results show all LLMs fall short of real humans, with GPT-4o leading.

**Compliance With Llm Reviewing Policy:**

Affirmed.

**Final Justification:**

I thank the authors for the additional experiments and clarifications in the second round.

Q1/Q3 (Metric validity) — largely resolved. The new cross-domain human evaluation on Amazon and Foursquare, together with instance-level Spearman correlations between human ratings and framework scores (0.52–0.61, p<0.001), directly addresses my earlier request for external validation. Inter-annotator agreement is also reasonable (Krippendorff's α 0.69–0.82). While the correlations are moderate rather than strong, they provide meaningful evidence that the metrics capture human-judged behavioral fidelity rather than purely encoder-specific similarity. I appreciate the effort to run new experiments across domains.

Q4 (Novelty) — acceptable. I already considered originality fair rather than poor. The authors' reframing of the contribution as a shift from task-level correctness to distributional measurement is reasonable, and I view this as a well-executed integration with a meaningful change in evaluation target.

Q5 (Failure mode analysis) — unresolved, and I believe my concern was misread. My point was not that the paper must provide mechanistic or architecture-level explanations. My concern is that the three "failure modes" (distributional averaging, weak persona conditioning, temporal inconsistency) are essentially rephrasings of the three low-scoring metrics (diversity, predictability, consistency), not analytical insights derived from them. Strong evaluation papers typically complement quantitative results with qualitative error analysis, case studies, or pattern discovery that go beyond restating which metric scored low. The authors' response defends the scope of the paper against a stronger claim (mechanistic interpretability) that I did not intend to make. **This concern remains.**


Nevertheless, The rebuttal made substantive methodological improvements on the most important issue I raised (external validation of the metrics), and I want to reward that effort. The remaining concerns on failure analysis depth are real but overall I am raising my score from 3 to 4 (Borderline accept).

**Key Questions For Authors:**

See weakness

**Limitations:**

yes

**Strengths And Weaknesses:**

Pros:
1. the core idea is reasonable and the motivation is clear
2. the paper is well written and easy to follow
3. authors show real empirical effort and also cover real0world datasets.
4. the human evaluation in appendix adds credibility
5. the ablation on encoder choice and classifier head is thorough


Cons:
1. I have some concerns with the metrics. I think it has some circular or fragile properties that aren't adequately addressed. 1) Distinguishability is measured as classifier error rate, but this is deeply dependent on how well- trained the classifier is and how separabale the embedding space is. A weak classifer would make all models look good; a strong one might overfit to superficial style differences. the paper doesn't justify why the trained encoder generalizes fairly across LLMs with very different output styels.
2. the world building consistency issue is telling: smaller models scored aritificaily high on consistency because they consistently write in AI style. The authors fix this with a Llama-3.1-8B rewriting step but using one LLM to post-process outputs before evaluating other LLMs introduces its own bias. Does the rewriter preserve semantic content faithfully? This isn't validated.
3. Predictability (Eq. 4–5) measures how well the user embedding predicts LLM-generated next actions. But the user encoder is trained on real human data — it's not obvious that similarity in this embedding space means the LLM is acting human-like, rather than that it happens to produce outputs that land near human behavior in a space designed for humans. The metric may be measuring something closer to recommendation system hit rate than genuine persona fidelity.
4. The framework is a reasonable engineering contribution, but conceptually it's not far from existing work on LLM-based user simulation in recommender systems. The novelty largely comes from combining existing components (sequential user encoders, distribution comparison, multi-domain application) rather than introducing a fundamentally new idea.
5. The finding that "LLMs fall short of real users" and "larger models are better" is not surprising. The paper doesn't offer much insight into why the gap exists or what the failure modes reveal about LLM architecture or training. The diversity collapse finding (t-SNE Figure 5) is the most interesting empirical result but is shallow right now.

---

> ### Author Rebuttal · Authors · 2026-03-31
>
> **Q1**: Our user encoder is trained on real human behavioral data, anchoring the representation space to meaningful human distributions. For Distinguishability, we train a separate classifier head for each LLM against real user data, avoiding cross-model bias and ensuring fair comparison.  Therefore, each LLM’s simulated behaviors are evaluated with a dedicated classifier against human data, rather than a shared classifier for all LLMs, eliminating concerns about cross-LLM generalization. Moreover, as shown in Appendix B.2, C.4–C.6, we conduct extensive hyperparameter tuning, ablations, case studies, and human evaluation. The consistent relative rankings and alignment with human judgments suggest that our training is sufficient and the evaluation is robust.
>
> **Q2**: A typo: we use Llama-3.1-70B, not 8B, for rewriting. To validate semantic preservation, we conduct a human study: a graduate student evaluates 300 rewritten samples (20 random cases per model in Table 1) using a 1–5 Likert scale for content fidelity (1: content missing; 2: largely missing; 3: moderately missing; 4: minor loss; 5: fully preserved). The average score is 4.96, with only 8 cases rated 4 and 2 cases rated 3. This indicates that rewriting preserves semantics well and introduces minimal bias.
>
> **Q3**: The concern arises if similarity is measured w.r.t. a global human space. However, our Predictability is user-conditioned, not global. The user encoder is trained on real human data to model how future behaviors follow from a given user profile, i.e., it learns the conditional structure p(behavior∣user). Thus, Predictability is analogous to perplexity in NLP or hit rate in recommendation, fundamentally a conditional modeling score. Generic, high-frequency “human-like” outputs tend to lie near the population center and are not particularly close to any individual user embedding, resulting in limited predictability. High scores are achieved only when the model captures user-specific behavioral patterns, aligning with conditional distribution induced by that user’s history. Therefore, when the encoder encodes whether behaviors match a persona, higher predictability directly indicates stronger persona-conditioned human-likeness, rather than mere proximity to a human-designed space.
>
> **Q4**: While user simulation is a well-established paradigm, our novelty lies in how it is evaluated. Prior work on LLM-based user simulation in RS typically evaluates with accuracy-based metrics and relies on limited human-annotated user studies. Existing human simulation benchmarks (e.g., SIMBENCH, CogBench) are also task-based with predefined correct answers. In contrast, we adopt user encoder as a judger to learn real human behavior distributions and evaluate human-likeness from multiple novel dimensions. Key distinctions:
>
>  (1) removes predefined correctness and evaluates alignment with empirical human distributions
>
>  (2) measures human-likeness at population- and sequence-level across multiple dimensions
>
>  (3) is scalable and generalizable across diverse open-ended tasks
>
> Therefore, our contribution is a new evaluation paradigm and an innovative use of existing components, leveraging large-scale web data for scalable, automated, distribution-based evaluation.
>
> （Also see Q4 of Reviewer bgde and Originality weakness of Reviewer oD8C）
>
> **Q5**: Beyond expected trends (e.g., larger models perform better), our analysis goes beyond simple leaderboard comparisons. As also noted as a strength by Reviewer bgde, we conduct extensive studies including sequence length effects, prompt position sensitivity, t-SNE representation collapse, user similarity curves, encoder and classifier ablations, case studies, and human evaluation, providing a comprehensive diagnosis of model behaviors.
>
> From these analyses, we identify three key failure modes:
>
> (1) Distributional averaging, where models collapse to high-frequency, central patterns, failing to capture the full diversity of human behavior;
>
> (2) Weak persona conditioning, where outputs are generic, weakly aligned with user-specific preferences, or even exhibit “AI-style” artifacts (limited rationality), indicating insufficient grounding in human behavioral logic;
>
> (3) Temporal inconsistency, where models struggle to maintain coherent, user-consistent behavior over long sequences.
>
> These findings suggest limitations in current LLM training. To build stronger human simulators, models should better capture preference differentiation across personas, support long-horizon behavioral consistency, and incorporate broader exploration of diverse human behaviors across domains, enabling more general and robust simulation capabilities. For example, agentic RL settings with multi-turn interaction, where rewards explicitly encourage consistency and user-conditioned predictability, could help models learn stable and personalized behavioral dynamics.
>
> We will elaborate on this part in the final version of our paper.

---

> > ### Author Rebuttal · Reviewer_DTjL · 2026-04-01
> >
> > I thank the authors for the thorough rebuttal.
> >
> > **Q2 (Rewriting bias)** is largely resolved — the human study on semantic preservation is convincing.
> >
> > **Q1 & Q3 (Distinguishability & Predictability)** are partially resolved. I appreciate the clarification on per-LLM classifier heads and user-conditioned predictability. However, consistent relative rankings demonstrate internal consistency of the pipeline, not that the metrics capture genuine behavioral human-likeness rather than stylistic or distributional fit. External validation (e.g., correlating metric scores with human judgments at the sequence level) would be needed to fully close this gap.
> >
> > **Q4 (Novelty)** remains my concern. The listed distinctions from prior work (removing predefined correctness, population-level evaluation, scalability) are meaningful but represent differences in scope and application. The core components — sequential user encoders, embedding-space classifiers, distribution comparison — are individually established. I view this as a competent integration rather than a conceptual advance, and a short rebuttal cannot easily change this framing. (I do not consider the originality to be poor, it's fair. I think it is a well-executed integration.)
> >
> > **Q5 (Shallow failure analysis)** is unresolved and weighs most heavily. The three failure modes (distributional averaging, weak persona conditioning, temporal inconsistency) are re-descriptions of low metric scores, not insights into why models fail. The authors acknowledge this will be elaborated in the final version, but this analysis is central to the paper's contribution and its absence significantly limits the current work's impact.
> >
> > I maintain my score at **3 (Weak Reject)**.

---

> > > ### Author Response · Authors · 2026-04-08
> > >
> > > **Q1/Q3**: we extend human evaluation beyond Worldbuilding and conduct human evaluation on Amazon/Foursquare. Following the same protocol, we recruite two graduate students specializing in societal AI, and keep them blinded to sequence source and model identity.
> > >
> > > For Rationality/Consistency, we use the same design as in Worldbuilding: for each dataset, we randomly sample 100 users and evaluate behavior sequences from three LLaMA models and real user. Annotators rate whether each sequence is plausible for the given user profile (Rationality) and whether it exhibit stable and coherent behavioral tendencies across actions (Consistency), using a 1–5 Likert scale. For Diversity, since Amazon/Foursquare contain fixed user-specific choice sequences rather than multiple users responding to a shared prompt, we use a group-level protocol: in each dataset, we randomly sample 150 users, divided them into groups of 10, and ask annotators to rate whether the sequences within each group exhibited diverse, distinctive, and non-template-like user behaviors, rather than collapsing to homogeneous or average-user-like patterns.
> > >
> > > The added human evaluations show trends consistent with our computational metrics. Inter-annotator agreement is also strong. Using ordinal Krippendorff’s α, we obtain 0.75/0.69/0.76 on Amazon and 0.72/0.79/0.82 on Foursquare for Rationality/Consistency/Diversity, respectively.
> > >
> > > We further examine whether framework scores align with human judgments at the instance level. For the same evaluated sequences, we compute the Spearman correlation between human ratings and the framework’s continuous logits for Rationality and Consistency (Diversity cannot be calculated). We observe positive correlations (p<0.001), indicating that sequences assigned higher scores by the framework are also judged by humans as more human-like. (On Worldbuilding, the correlations are 0.55 and 0.59 for Rationality and Consistency, respectively)
> > >
> > > Overall, these new results provide direct cross-domain external validation, supporting that our metrics capture meaningful human-likeness rather than merely encoder-specific similarity, and strengthen the generality of the framework beyond the original Worldbuilding setting.
> > >
> > > |Amazon|Real User|Llama-3.1-8B|Llama-3.1-70B|Llama-3.1-405B|Spearman|
> > > |-|-|-|-|-|-|
> > > |Rationality|4.20|1.12|2.35|2.76|0.58|
> > > |Consistency|4.42|3.10|3.94|4.11|0.52|
> > > |Diversity|4.45|2.96|3.40|3.72|--|
> > >
> > > |Foursquare|Real User|Llama-3.1-8B|Llama-3.1-70B|Llama-3.1-405B|Spearman|
> > > |-|-|-|-|-|-|
> > > |Rationality|4.52|2.10|2.96|3.22|0.55|
> > > |Consistency|4.36|1.22|3.05|3.96|0.61|
> > > |Diversity|4.28|3.14|3.56|3.88|--|
> > >
> > > **Q4**: Thank you for recognizing the work as a well-executed integration. Our novelty does not lie in introducing new individual building blocks such as sequential encoders. Our claim is instead that the conceptual contribution lies in the formulation of evaluation problem. Prior work typically evaluates human simulation fidelity at the level of predefined tasks, correctness-level evaluation. But we formulate it as a distributional measurement problem: whether a model reproduces the population-level and sequence-level structure of real human behavior across users, especially in open-ended settings without a single ground truth. Thus, the contribution is not merely assembling existing tools, but operationalizing a different evaluation target that prior task-level formulations do not directly capture. We agree this is not a new backbone method, but we believe this change in what is measured—and how simulation fidelity is defined—is a meaningful conceptual advance beyond scope or application alone.
> > >
> > > **Q5**: We agree that the current paper does not provide a mechanistic explanation of why these weaknesses arise at the level of model internals/architecture. But this is also not the intended scope of the work. Our contribution is not a mechanistic interpretability study, but a framework/benchmark for discovering, quantifying, and validating where current LLMs fall short in human behavior simulation. A core contribution of an evaluation paper is to make previously under-specified weaknesses measurable and visible. In this sense, our goal is to turn the broad claim that “LLMs are not yet human-like” into quantifiable, cross-domain, multi-dimensional empirical failure patterns. The patterns discussed in Q5 — distributional averaging, weak persona conditioning, and temporal inconsistency — are therefore not intended as neuron-level or architecture-level causal explanations, but as the main weaknesses consistently revealed by the framework across domains and analyses. We agree that deeper explanations of their root causes are important, but without a reliable evaluation framework that identifies what the weaknesses are and where they manifest, such mechanistic analysis is difficult to ground. We will clarify in final version that deeper mechanistic analysis is an important direction for future work rather than a claim of present paper

---

### Official Review · Reviewer_oD8C · 2026-03-13

**Soundness:** 3
**Presentation:** 3
**Significance:** 3
**Originality:** 2
**Overall Recommendation:** 4
**Confidence:** 5

**Summary:**

# RQ
How can we move beyond the traditional Turing Test to evaluatе AI "human-likeness" at a population/distribution level rather than just through short, individual conversations?

# Contribution:
- Automated framework to evaluate human-likeness using public data
- Multi-dimensional metrics covering rationality, consistency, and diversity to transition from evaluating single-step tasks to long-term behavioral sequences.
- conducted extensive tests on mainstream LLMs (including GPT-4o, Llama-3.1, and Qwen), revealing that while top models show promise, a significant "human-likeness gap" remains

**Compliance With Llm Reviewing Policy:**

Affirmed.

**Key Questions For Authors:**

1. Evidence of data decontamination or a "temporal split" analysis (evaluating on data post-dating the models' knowledge cutoffs
2. Did you conduct a sensitivity analysis to see how much the results change with different prompt templates? For instance, if a "Chain-of-Thought" prompt or a different persona description was used, would the "human-likeness gap" close or widen?
3. Have you validated the User Encoders against human-labeled behavioral traits (e.g., Big Five personality scores) to ensure they are capturing meaningful psychological/behavioral signatures rather than just statistical surface patterns?
4. In Table 1, some models show high diversity but lower predictability (or vice-versa). Is there an inherent trade-off in your framework where a model might be penalized for being "too diverse" (appearing random) or "too predictable" (appearing robotic)?
5. You used Llama-3.1-8B in Appendix C.1 to "rewrite" answers to remove "AI tone" for the Consistency metric. Does this rewriting process potentially strip away the very behavioral nuances (tone, style, idiosyncratic errors) that define a specific human user?

**Limitations:**

Yes

**Strengths And Weaknesses:**

# Soundness
## Strength
- 3 datasets (Amazon, StackExchange, Foursquare) for 3 domains (shopping, Q&A, mobility).
- Rationality, Consistency, and Diversity metrics
- automated evaluator using trained user encoders (SASRec, e5-base-v2) and classifiers
- ablation studies on classification heads and a blinded human evaluation (with high inter-annotator agreement)
## Weakness
- historical web data as "ground truth" human behavior
  - Web data (like StackExchange) is often curated or performative and may not represent authentic daily human behavior.
- "LLM Simulator" relies heavily on hand-engineered prompts, no prompt sensitivity ablation
- cosine similarity in high-dimensional embedding spaces to measure "Predictability" and "Diversity ==> may overlook subtle but critical qualitative differences in human logic
- does not test how LLMs adapt their "human-likeness" in response to dynamic, unpredictable human feedback
- All 3 datasets are old, so all these large models may have already been trained on them already

# Presentation
## strength
- flows, effective visuals, good appendix
## weakness
- Table 1 is bit dense
- Human-likeness is used somewhat interchangeably with believability and rationality
- lacks a deep qualitative discussion on why certain models (like Qwen vs. Llama) exhibit specific behavioral biases beyond general scaling laws.
- radar chart is slightly misleading, normalization helps comparison but masks the absolute magnitude of the human-likeness gap,

# Significance
## stregnth
- behavioral distribution by defining rationality, consistency, and diversity
- cross-domain validation
- quantifies exactly how much current top-tier models (like GPT-4o) still deviate from real human patterns
## weakness
- representation bias (western demographics)
- historical web data (static data, possible data leakage)

# Originality
- behavioral distribution, multi-domain benchmark
## weakness
- idea isn't too new (e.g., https://arxiv.org/html/2509.21501v1, next action prediction task (https://arxiv.org/abs/2603.05923), ) 1.

---

> ### Author Rebuttal · Authors · 2026-03-31
>
> **Q1**: Web data may reflect platform-specific or performative aspects of behavior. However, our goal is to capture observed behavioral traces in real-world platforms, where such factors are also part of human behavior. For data leakage, while individual instances may overlap with pretraining data, our evaluation focuses on user-conditioned behavioral sequences rather than isolated cases, which substantially reduces memorization risk. e.g., in Worldbuilding, each question is associated with multiple users and diverse responses; generating appropriate behavior requires modeling user profiles, not recalling any single instance. Moreover, our large-scale, user-specific trajectories are unlikely to be memorized as complete sequences, making the evaluation effectively novel even if isolated cases were seen.
>
> **Q2**: Our prompts are carefully designed rather than ad hoc. We iteratively refine their wording and structure through pilot experiments to ensure stable instruction-following behavior across LLMs. We conduct an additional analysis using CoT prompting. The results show only slight/no gains across all three datasets, indicating that our prompt design is already well-optimized and robust. We report Amazon results here, full results will be provided during discussion.
> || Cons|Dist|Pred|Div
> |-|-|-|-|-
> |Qwen2.5-3B|0.2120|0.1839|0.0005|0.5372
> |Qwen2.5-14B|0.7235|0.1841|0.1026|0.5812
> |Qwen2.5-72B|0.8169|0.2402|0.1375|0.6110
>
> **Q3**: While we can not explicitly align embeddings with predefined psychological traits (e.g., Big Five), user encoders are trained on large-scale human behavioral data, enabling them to capture both fine-grained distinctions and meaningful behavioral patterns beyond surface statistics. We provide empirical validation: Case study (Figures 7, 13–15) show that metric variations correspond to interpretable behavioral differences across datasets, and human evaluation in Appendix C.6 further confirms that our metrics align with human judgments. These results suggest that the learned representations capture meaningful behavioral signatures, rather than merely superficial patterns.
>
> **Q4**: There is no trade-off. Both metrics are measured relative to real human behavior distributions, rather than penalizing either property itself. When a model accurately captures specific behaviors of each user, it can achieve high scores on both metrics (e.g., GPT-4o performs well). Moreover, random variation does not improve diversity (as shown by our random baseline in Reviewer bgde Q3), since our metric rewards alignment with human population diversity, not arbitrary randomness. The apparent trade-off arises from model limitations. By qualitative analysis, we observe certain model-specific errors. e.g, In Worldbuilding, for a question such as “how to design a society on a resource-scarce planet,” Qwen tends to generate high-frequency, central-pattern responses (e.g., generic resource allocation and hierarchical structures), leading to high predictability but low diversity. In contrast, Llama produces more varied answers (e.g., physics-driven vs. political vs. narrative designs), increasing diversity, but sometimes misaligns with specific user styles, resulting in lower predictability.
>
> **Q5**: See reply to Reviewer bgde's Q5
>
> **Soundness**: By “dynamic, unpredictable feedback” we understand this as interactive settings where model behavior adapts to human responses over time. Our current framework focuses on evaluating behavioral simulation from historical traces. So we do not model closed-loop interaction with feedback. This aspect is not our focus, and we will discuss it in limitation section in final version.
>
> **Presentation**: Will improve readability of Table1 by reorganizing the layout. For radar chart, absolute values make the scale too large to draw. So we adopt normalization to facilitate comparison across dimensions. Absolute values are fully reported in Table1, serving as a complementary reference.  Human-likeness/believability is the overall evaluation target, while rationality is one of the specific dimensions used to assess it. We do not intend to use these terms interchangeably. In the revision, we will clarify this distinction and ensure consistent terminology.
>
> **Significance**: The framework evaluates human-likeness w.r.t the chosen population. English is used as the most common setting; multi-language/region extensions are not our focus and future work.
>
> **Originality**: While related, these works differ in objective and scale. The first focuses on domain-specific simulation with small, human-annotated user study, and the second formulates behavior modeling as next action prediction on limited user trajectories. But our goal is distributional evaluation of human-likeness, not prediction accuracy. Our framework leverages large-scale behavioral data, operates at the population level, and scales across domains beyond manually curated, small-user settings. Refer to Reviewer bgde Q4 for more.

---

> > ### Author Rebuttal · Reviewer_oD8C · 2026-04-03
> >
> > Thanks. I'll maintain my score.
> > Q2 is answered.
> >
> > Q1 doesn't seem easy to be resolved.
> > Q5 isn't resolved.

---

> > > ### Author Response · Authors · 2026-04-08
> > >
> > > **Q1**: We agree that strict decontamination or a clean temporal-split guarantee is difficult to fully establish for large-scale web data. However, our evaluation does not rely on the assumption that every underlying raw instance was unseen during pretraining. The key point is that our task is not isolated instance reproduction, but user-conditioned sequential simulation. We reorganize the data into persona-based behavior sequences, where the model must infer a user’s profile from the first half of the history and simulate the second half step by step. Seeing scattered raw data points during pretraining would not allow the model to reconstruct these sequence-level, user-conditioned simulations directly.
> > >
> > > As can be seen from our experimental results, including both Table 1 and the human evaluation （Appendix C.5, C.6）, current LLM performance is still far from satisfactory. If severe data leakage were truly occurring, the models’ performance should be much closer to that of real humans.
> > >
> > > **Q3**: Please refer to Q1 in our Reply Rebuttal Comment to Reviewer bgde. We additionally conducted human evaluations on the other two datasets and analyzed the Spearman correlation between human ratings and our framework scores. The results demonstrate that our metrics capture meaningful human-likeness rather than merely statistical surface patterns.
> > >
> > > **Q5**: In open-ended writing settings, surface-level style and deeper behavioral patterns jointly constitute human behavior. For all metrics except Consistency, we train the encoder on human behaviors that include both aspects. Therefore, we do not believe the metric/framework itself should be invariant to tone/style; these are inherently part of behavior. Only for Consistency do we observe that the current tone/style deficiencies of LLMs interfere with the judgment: although LLM simulations may be less consistent in deeper viewpoints, they often remain highly consistent in AI-style tone. These two factors can conflict. As the original encoder considers both aspects simultaneously, stylistic consistency can distort the accuracy of the metric. This is not a problem of the framework, but rather a shortcut caused by limitations of current LLMs. Therefore, removing the influence of tone/style and evaluating Consistency only on deeper behavioral patterns can more meaningfully reflect the current performance of LLM simulation in open-ended writing settings. This issue does not arise in the other two domains.
> > >
> > > Our response to Reviewer DTjL’s Q2 also conducted human evaluation to show that the rewriting process itself has high fidelity and introduces minimal bias.

---

### Official Review · Reviewer_FzK8 · 2026-03-13

**Soundness:** 3
**Presentation:** 4
**Significance:** 3
**Originality:** 3
**Overall Recommendation:** 5
**Confidence:** 3

**Summary:**

The authors propose a computational framework for evaluating human-likeness in LLMs' open-ended human behaviors, moving beyond simple QA-based evaluation. They consider three experimental scenarios: online shopping, open-topic Q&A, and Foursquare for urban mobility to assess the alignment between major existing LLMs and real human open-ended behavior, revealing that LLMs still exhibit a significant gap from real user behavior.

**Compliance With Llm Reviewing Policy:**

Affirmed.

**Final Justification:**

My concern have been resolved.  I hope all of these points will be incorporated into the revised version of the paper.

**Key Questions For Authors:**

- As shown in Table 2, the authors considered different User Encoder models. I would like to know what additional measures are taken to ensure the reliability of the User Encoder. Could the authors provide a more specific (and broader) explanation and elaboration on this point?

- Is there a certain correlation between the Consistency and Predictability/Distinguishability metrics? （overlap)

- From Table 1, the values of the Dist, Pred, and Div metrics are all relatively low. Could the authors provide an error analysis or explain the deeper, specific reasons behind these low values (from the perspective of the overall experimental dataset)?

**Limitations:**

Human Behavior is extremely broad. Although this paper proposes a computational framework, there remains a gap if one aims to evaluate human-likeness in a complete, comprehensive, and all-encompassing manner (covering all scenarios). The authors could acknowledge this point in the limitations section.

**Strengths And Weaknesses:**

- The paper is clearly written, well-structured, and highly readable.
- The research question is valuable and meaningful. The evaluation approach differs from previous single-turn or QA-only assessments by adopting the perspective of open-ended behavioral sequences, and the proposed framework demonstrates good scalability.


- The reliability of the experimental conclusions hinges on the effectiveness of the User Encoder. The authors are advised to provide additional experiments to further validate the User Encoder.
- The paper lacks more in-depth discussion and analysis of certain metric values.

---

> ### Author Rebuttal · Authors · 2026-03-31
>
> **Q1**: We take several steps to ensure the reliability of the user encoder.
>
> 1. we conduct robustness analysis across multiple encoder architectures and observe consistent relative rankings (see Tables 1, 6, 7), indicating that conclusions are not tied to a specific model.
> 2. the encoders we adopt are widely used, high-capacity open-source models, ensuring sufficient representation power for behavioral modeling.
> 3. the encoder is trained on real human behavioral data, anchoring the representation space to meaningful human distributions, and we perform dataset-specific hyperparameter tuning (Appendix B.2) to optimize performance.
> 4. we provide post-hoc validation: our evaluation results align with human judgments, and case studies further confirm that the encoder captures meaningful behavioral differences.
>
> Together, these steps ensure that the encoder provides a stable and reliable basis for evaluation.
>
> **Q2**: These metrics capture complementary aspects of human-like behavior. Distinguishability measures whether human and model behaviors are separable at the distribution level; predictability evaluates whether the model follows user-specific behavioral patterns; and consistency examines whether a simulated sequence maintains coherent behavior for the same user over time.
> A typical case arises when the model exhibits slight instability in long sequences. For example, in Amazon, the simulated sequence may largely follow the user’s historical preference (e.g., serious sci-fi), making the overall sequence predictable from the user embedding. However, if the model occasionally produces out-of-pattern choices (e.g., a few popular action movies), these deviations can break the internal coherence between segments, leading to lower consistency.
> Thus, predictability reflects alignment with user-conditioned distributions, while consistency captures stability of behavior over time, and the two are not redundant.
>
> **Q3**: The relatively low values are expected and reflect both the nature of the task and dataset characteristics.
> For Distinguishability, the metric is based on classifier error rate. We observe that human behaviors are relatively hard to distinguish among themselves (an error rate of 50% is as expected), indicating that the learned space is well-grounded. In contrast, human vs. LLM behaviors remain separable at the population level, suggesting that current models still fail to fully capture human behavioral patterns.
> For Predictability and Diversity, both are computed via embedding similarity and thus depend on the encoder and data modality. In Amazon, we use an ID-based sequential model (SASRec), where prediction is inherently difficult, leading to lower absolute values (consistent with recommender system literature). In contrast, for Worldbuilding and Foursquare, text-based embeddings with higher token overlap yield higher similarity scores.
> Therefore, we emphasize relative comparisons under a fixed setting, while absolute values naturally depend on the encoder and dataset properties.
>
> **Limitations**: We thank the reviewers for the reminder. In Section 2.6, we have provided some clarifications regarding the scenarios in which this framework is applicable. One possible limitation is that our framework currently simulates and evaluates only a single type of behavior at a time. In real-world scenarios, user behavior is often multi-faceted. For example, on Twitter, users engage in various actions such as retweeting, liking, and posting, all of which are interdependent. However, our framework is designed to evaluate behaviors in isolation (e.g., focusing solely on posting behavior), which may oversimplify user interaction dynamics. Two key challenges remain: (1) enabling LLMs to simulate multiple behaviors simultaneously in a cohesive manner, and (2) accurately modeling the complex behavior patterns. Addressing these challenges would be a significant step toward improving the authenticity and generalizability of LLM-based behavior simulation and simulation evaluation. We will include a "Limitations" section in the final version.

---

> > ### Author Rebuttal · Reviewer_FzK8 · 2026-04-03
> >
> > Thank you for the detailed response. I have raised my score.

---

> > > ### Author Response · Authors · 2026-04-08
> > >
> > > Thank you for your thoughtful consideration and for taking the time to re-evaluate our work. We sincerely appreciate your recognition.

---

### Official Review · Reviewer_bgde · 2026-03-22

**Soundness:** 2
**Presentation:** 2
**Significance:** 2
**Originality:** 2
**Overall Recommendation:** 3
**Confidence:** 4

**Summary:**

The paper proposes a distribution-based framework for evaluating how human-like LLM-generated behaviors are in open-ended, sequential settings. Its central claim is that existing evaluation practices such as short conversational Turing-style tests and task-specific benchmarks do not adequately capture the distributional and temporal structure of real human behavior, so the authors instead construct a framework around large-scale web behavior logs. They evaluate on three domains—Amazon Movies&TV, StackExchange Worldbuilding, and Foursquare check-ins—using a simulator, a learned user encoder, and an evaluator that measures rationality, consistency, and diversity. Across models, performance improves with scale, but the best system still remains meaningfully below real users, with GPT-4o leading yet still trailing human behavior on average.

**Compliance With Llm Reviewing Policy:**

Affirmed.

**Final Justification:**

Based on authors' response, I have updated my scores. My main concern is still a lack of empirical evaluation which needs one more revision.

**Key Questions For Authors:**

Please check the strengths and weaknesses.

**Limitations:**

yes

**Strengths And Weaknesses:**

Strengths:
1. The pipeline is coherent and easy to understand: behavior dataset → LLM simulator → user encoder → evaluator. That modularity makes the paper intellectually clean, and the three-way decomposition of human-likeness into rationality, consistency, and diversity gives the framework an interpretable structure that is more informative than a single aggregate score. I also like that the authors explicitly target both micro-level behavior fidelity and macro-level population diversity, which is a sensible fit for social simulation and human behavior modeling.

2. The paper compares many model families and sizes, shows a clear scaling trend, and demonstrates that different families have different strengths: for example, Qwen2.5 is relatively stronger on instant behavior-related metrics, while Llama-3.1 tends to be better on stability and diversity. The broad comparison across Amazon, Worldbuilding, and Foursquare makes the result more convincing than a single-domain benchmark would be, and the authors also show that Worldbuilding is the hardest setting, which is a useful and nontrivial finding.


3. I also think the paper is stronger than average in its analysis section. It does not stop at a leaderboard: it studies sequence length effects, prompt-position sensitivity, t-SNE collapse, user-similarity curves, encoder ablations, classifier-head ablations, and a human evaluation with blinded annotators. That makes the paper feel more like a framework paper than a one-off benchmark report. The human study is especially helpful because it qualitatively supports the same ordering seen in the automated metrics, with substantial inter-annotator agreement.




Weaknesses:
1. The main weakness is that the entire evaluation stack depends on a learned representation space and learned downstream classifiers. Distinguishability is a classifier-based human-vs-model test, consistency is another classifier on paired sequence segments, and diversity is computed through nearest-neighbor geometry in embedding space. That means the final scores are only as trustworthy as the encoder, embedding geometry, and classification heads. The ablations show some robustness in relative ranking, but the absolute values do move around, which reinforces the concern that the metrics are proxy-dependent rather than direct measurements of human-likeness.


2. The paper does not include an ablation or comparative analysis evaluating how the proposed metrics behave without reliance on the encoder (e.g., directly in the original prompt, action, or retrieval space). This is particularly important given that alternative evaluation paradigms exist, such as direct model-based evaluation frameworks (e.g., LLM-as-a-Judge), which do not strictly rely on a fixed embedding encoder


3. A random or trivial baseline is missing, which is important for calibrating the scale and interpretability of the reported metrics. Without such baselines, it is difficult to determine whether the observed scores meaningfully exceed simple heuristics or reflect artifacts of the evaluation setup. Currently, only the upper bound (real user performance) is reported.

4. The novelty and positioning of the proposed framework relative to existing work on evaluating LLM-based human simulation could be further clarified. Recent works such as SIMBENCH: Benchmarking the Ability of Large Language Models to Simulate Human Behaviors, How Far are LLMs from Being Our Digital Twins?, CogBench: A Large Language Model Walks into a Psychology Lab, and Beyond Believability: Accurate Human Behavior Simulation with Fine-Tuned LLMs explicitly evaluate simulation fidelity using behavioral, cognitive, or persona-based benchmarks. These approaches assess whether models can replicate human decisions, preferences, and behavioral patterns in realistic settings. In contrast, the proposed framework relies on embedding-based distributional similarity, and the paper would benefit from a clearer discussion of how this approach complements or improves upon prior simulation-oriented evaluation methods.

5. A related issue is surface-form sensitivity. The appendix explicitly notes that in Worldbuilding, the authors observed an “AI tone” in generated answers and then rewrote responses with Llama-3.1-8B to remove style and formatting effects before reporting the main results. That is a strong sign that the metric can be affected by stylistic artifacts, not just deeper behavioral fidelity. In other words, the framework is partly measuring whether outputs look human in the embedding space induced by the chosen encoder, which is informative but also somewhat fragile.

---

> ### Author Rebuttal · Authors · 2026-03-31
>
> **Q1**: Our metrics are representation-based and rely on a learned encoder. This is by design, as human-likeness in open-ended behaviors does not admit a direct observable scalar metric. Any distribution-level comparison between human and model behaviors necessarily requires a shared representation space; thus, some form of learned representation is unavoidable rather than a limitation specific to our approach. This is analogous to widely used metrics such as FID, which are also representation-dependent but effective in capturing high-level semantic structure. Our goal is not to define an absolute measure of human-likeness, but to provide a consistent comparative framework across models. Empirically, we show that conclusions are not tied to a specific encoder: relative rankings are stable across encoder/classifier variants, and importantly, align with human evaluation. In addition, the encoder is trained on real human behavioral data, anchoring the space to meaningful human distributions rather than arbitrary features.
> Therefore, while proxy-based, our metrics are a principled and necessary design choice for distributional behavior comparison, and yield robust, human-aligned comparative insights.
>
> **Q2**: We agree that encoder-free evaluation is valuable, but no existing method fairly matches our distribution-level setting. As noted in Appendix C.6, our human evaluation protocol can be adapted to LLM-as-a-Judge (only for Worldbuilding).
> However, judge-based methods operate at instance level with prompt-dependent scores. In contrast, our framework uses a shared representation space for user-conditioned and distribution-level analysis. Diversity further captures distributional properties (e.g., coverage, mode collapse) that instance-level scoring cannot access. Moreover, for datasets like Amazon and Foursquare, LLM judges may lack reliable knowledge of items, time, or spatial constraints, making even instance-level evaluation noisy. Extending such methods to population-level properties (e.g., diversity across many users) is infeasible due to context and scalability limits. Thus, LLM-as-a-Judge is complementary, while our method enables scalable, structured evaluation of behavior distributions.
>
> **Q3**: We add a random baseline that removes persona: we prompt Qwen-2.5-72B without user profile to answer question or select item. Results show a substantial drop across all metrics. This indicates that capturing user-specific information is essential for achieving human-like behavior under our framework.
> | Domain        | Cons  | Dist  | Pred   | Div   | Avg   |
> |---------------|-------|-------|--------|-------|-------|
> | Amazon        | 0.1550| 0.0720| -0.0520| 0.4849| 0.1649|
> | Worldbuilding | 0.6322| 0.0050| 0.6924 | 0.6240| 0.4884|
> | Foursquare    | 0.1035| 0.0122| 0.7154 | 0.8465| 0.4194|
>
> **Q4**: Our framework differs in evaluation target and granularity, and is complementary to prior approaches. Existing methods (SIMBENCH, BehaviorChain, CogBench, Beyond Believability) primarily adopt task-based evaluation, assessing whether models can reproduce human decisions or actions in predefined settings with explicit notions of correct answers. But our framework does not assume predefined correctness. Instead, it models real human behavior distributions directly from data, and evaluates whether LLMs reproduce the distributional structure of these behaviors in open-ended settings. This leads to three key distinctions:
>
>  (1) No predefined correct answers: evaluation is based on alignment with empirical human distributions, rather than task-specific accuracy.
>
>  (2) Population- and sequence-level evaluation: we measure human-likeness from multiple dimensions (e.g., diversity, consistency) across many users and behavior sequences, rather than reducing evaluation to whether individual outputs match a single correct answer.
>
>  (3) Scalability and generality: our framework is data-driven and can be applied to diverse domains without redesigning tasks, whereas prior work relies on fixed scenarios and evaluation protocols that are difficult to transfer across settings.
>
> The embedding space serves as a tool to enable such distributional comparison. Therefore, our approach complements prior work by enabling scalable, generalizable evaluation of human behavior distributions beyond correctness-based benchmarks.
>
> **Q5**: For text generation tasks, both surface-level style and deeper behavioral patterns are natural components of human-likeness, and our framework retains both in most metrics. The exception is consistency: due to model limitations, a uniform “AI style” can act as a shortcut, yielding artificially high scores without true behavioral coherence. We therefore apply rewriting only for this metric to remove such surface-form bias and focus on deeper consistency. This issue is specific to open-ended text (e.g., Worldbuilding); in structured domains like Amazon and Foursquare, stylistic artifacts do not arise.

---

> > ### Author Rebuttal · Reviewer_bgde · 2026-04-06
> >
> > The rebuttal is generally strong in clarification and positioning, and it partially addresses several of my concerns. However, it does not fully resolve the core methodological issue regarding representation dependence and metric validity. Below I reassess each of my points.
> >
> > Q1 -> Representation dependence (core concern) -  The framework is entirely dependent on learned embeddings and classifiers, making the metrics proxy-based and potentially fragile. I agree that some shared representation is necessary for distributional comparison.  However, the core issue is only partially resolved: The analogy to FID is appropriate but also highlights known limitations (sensitivity to representation choice).
> > Stability of rankings does not guarantee that the metric captures true human-likeness rather than encoder-induced similarity.
> > Human evaluation evidence is limited in scope (primarily one domain), so it does not fully validate the generality of the framework.
> >
> > Q4 -> The argument is mostly conceptual, not empirical. There is still no direct comparison or bridging experiment
> >
> > Q5 -> The need for rewriting suggests the metric is not invariant to surface form. The fix introduces an additional processing step that may itself bias results. It highlights that the framework can conflate style with behavior
> >
> > I can change my score depending on the answer to these questions.

---

> > > ### Author Response · Authors · 2026-04-08
> > >
> > > **Q1**: we extend human evaluation beyond Worldbuilding and conduct human evaluation on Amazon/Foursquare. Following the same protocol, we recruite two graduate students specializing in societal AI, and keep them blinded to sequence source and model identity.
> > >
> > > For Rationality/Consistency, we use the same design as in Worldbuilding: for each dataset, we randomly sample 100 users and evaluate behavior sequences from three LLaMA models and real user. Annotators rate whether each sequence is plausible for the given user profile (Rationality) and whether it exhibit stable and coherent behavioral tendencies across actions (Consistency), using a 1–5 Likert scale. For Diversity, since Amazon/Foursquare contain fixed user-specific choice sequences rather than multiple users responding to a shared prompt, we use a group-level protocol: in each dataset, we randomly sample 150 users, divided them into groups of 10, and ask annotators to rate whether the sequences within each group exhibited diverse, distinctive, and non-template-like user behaviors, rather than collapsing to homogeneous or average-user-like patterns.
> > >
> > > The added human evaluations show trends consistent with our computational metrics. Inter-annotator agreement is also strong. Using ordinal Krippendorff’s α, we obtain 0.75/0.69/0.76 on Amazon and 0.72/0.79/0.82 on Foursquare for Rationality/Consistency/Diversity, respectively.
> > >
> > > We further examine whether framework scores align with human judgments at the instance level. For the same evaluated sequences, we compute the Spearman correlation between human ratings and the framework’s continuous logits for Rationality and Consistency (Diversity cannot be calculated). We observe positive correlations (p<0.001), indicating that sequences assigned higher scores by the framework are also judged by humans as more human-like. (On Worldbuilding, the correlations are 0.55 and 0.59 for Rationality and Consistency, respectively)
> > >
> > > Overall, these new results provide direct cross-domain external validation, supporting that our metrics capture meaningful human-likeness rather than merely encoder-specific similarity, and strengthen the generality of the framework beyond the original Worldbuilding setting.
> > >
> > > |Amazon|Real User|Llama-3.1-8B|Llama-3.1-70B|Llama-3.1-405B|Spearman|
> > > |-|-|-|-|-|-|
> > > |Rationality|4.20|1.12|2.35|2.76|0.58|
> > > |Consistency|4.42|3.10|3.94|4.11|0.52|
> > > |Diversity|4.45|2.96|3.40|3.72|--|
> > >
> > > |Foursquare|Real User|Llama-3.1-8B|Llama-3.1-70B|Llama-3.1-405B|Spearman|
> > > |-|-|-|-|-|-|
> > > |Rationality|4.52|2.10|2.96|3.22|0.55|
> > > |Consistency|4.36|1.22|3.05|3.96|0.61|
> > > |Diversity|4.28|3.14|3.56|3.88|--|
> > >
> > > **Q4**: Direct empirical comparison is difficult because prior work and ours evaluate different targets. Prior benchmarks mainly ask whether a model can reproduce an expected decision/response in a predefined scenario, while our framework asks whether it reproduces the distributional structure of human behavior across users and sequences, especially in open-ended settings without a single ground truth.
> > >
> > > Our initial response to Q4 has already explained the novelty. In addition, we further emphasize two points. 1.In settings such as Worldbuilding, where there is no single correct answer, prior work cannot be readily applied through correctness-based evaluation. 2.Our multi-dimensional evaluation provides a finer-grained view than a simple correct/incorrect judgment. As shown in our experiments, different model families exhibit different strengths across dimensions—e.g., LLaMA models perform better in consistency and diversity, while Qwen2.5 models achieve stronger rationality. Such distinctions are difficult to observe with prior evaluation methods.
> > >
> > > **Q5**: In open-ended writing settings, surface-level style and deeper behavioral patterns jointly constitute human behavior. For all metrics except Consistency, we train the encoder on human behaviors that include both aspects. Therefore, we do not believe the metric/framework itself should be invariant to tone/style; these are inherently part of behavior. Only for Consistency do we observe that the current tone/style deficiencies of LLMs interfere with the judgment: although LLM simulations may be less consistent in deeper viewpoints, they often remain highly consistent in AI-style tone. These two factors can conflict. As the original encoder considers both aspects simultaneously, stylistic consistency can distort the accuracy of the metric. This is not a problem of the framework, but rather a shortcut caused by limitations of current LLMs. Therefore, removing the influence of tone/style and evaluating Consistency only on deeper behavioral patterns can more meaningfully reflect the current performance of LLM simulation in open-ended writing settings. This issue does not arise in the other two domains.
> > >
> > > Our response to Reviewer DTjL’s Q2 also conducted human evaluation to show that the rewriting process itself has high fidelity and introduces minimal bias.

---

### Decision · Program_Chairs · 2026-04-30

**Decision:**

Accept (regular)

**Comment:**

This paper introduces a distribution-based framework for evaluating human-likeness and believability of LLM behaviors using large-scale human behavior data on the web.

Summary of Contributions.
- Reviewers mostly agree that the problem is important and the paper is clearly written, with a clean and fairly comprehensive pipeline.
- Contributions include the design of metrics (rationality, consistency, diversity), cross-domain eval, and relatively thorough empirical analysis.
- The added human evaluations in the rebuttal (across domains, with reasonable agreement and moderate correlation to metrics) helped address concerns about validity.

Summary of Concerns
- The main issue, raised by multiple reviewers, is the heavy dependence on learned representations and classifiers, making the metrics somewhat proxy-dependent and fragile. The authors argue this is unavoidable for a distribution-level measurement, and further provide robustness checks in the rebuttal. Relatedly, sensitivity to surface form (e.g., removing "AI-tone" before eval) and lack of encoder-free or alternative evaluation comparisons challenge the reliability of the metrics.
- There's also a recurring concern about limited novelty or unclear positioning compared to existing work on evaluating LLM-based human simulation. It's conceptually quite similar to existing efforts, except for the embedding-based distributional similarity measure. It'd be good to add a clear discussion.
- Another concern includes potential data issues (e.g., data bias, potential data leakage).

Overall the author's rebuttal partially addressed some concerns, especially on cross-domain external validation. However, the concern about the reliability of the representation-based metrics may not have been fully resolved.